# Analysis and Testing of Rigid–Flexible Coupling Collision Harvesting Processes in Blueberry Plants

Haibin Wang *, Xiaomeng Lv, Feng Xiao  and Liangliang Sun

College of Engineering and Technology, Northeast Forestry University, 26 Hexing Road, Harbin 150040, China
* Correspondence: whb_nefu@nefu.edu.cn

**Abstract:** China possesses a vast territory, and the manual harvesting of blueberries is time-consuming and labor-intensive. Due to the planting agronomy differences in other countries, China needs to develop a domestic blueberry harvester to realize mechanical blueberry harvesting. In the harvesting process, "collision" is the core problem. Most of the literature has studied rigid body–rigid body collision, while few authors have studied rigid–flexible coupling collision mechanisms in the field of berry harvesting. In this paper, a rigid–flexible coupling collision model between the harvester and the blueberry plant was established based on the L-N nonlinear spring damping model (describing the collision force model between two colliding objects, consisting of the nonlinear spring and the damper) and improved the Coulomb model (the tangential collision force model), and the collision mechanism of blueberry harvesting was analyzed. The harvesting collision process was analyzed using both MATLAB and ADAMS software and the same conclusions were obtained: the collision force and fruit harvesting force were inversely proportional to the machine velocity but positively proportional to the rotational velocity of the hydraulic motor of the harvesting device. The following machine parameters were required to meet harvesting conditions: a harvesting device output rotational velocity of 120–150 r/min and a machine velocity of 40–50 m/min. A harvesting field test using a self-propelled blueberry harvester was conducted, which showed that the test results were consistent with the software simulation conclusions. When the machine velocity of the harvester and the output rotational velocity of the hydraulic motor were 45 m/min and 130 r/min, respectively, the machine provided optimum harvesting efficiency and fruit quality with the following optimum parameters: a harvesting efficiency of 5.1 kg/min, a raw fruit harvesting rate of 2.9%, and a damaged fruit harvesting rate of 3.6%. This research can lay the preliminary theoretical foundation for the analysis of a blueberry harvesting mechanism, and the research results can provide a theoretical reference for the harvesting of other similar berry shrubs.

**Keywords:** blueberry harvester; harvesting collision model; analysis of the collision process; harvesting field test; working parameters of harvester



## 1. Introduction

The blueberry is sought by the public for their delicate and sweet flesh and the many nutritional values they contain. However, in the process of harvesting blueberries, it has been found that simple harvesting equipment tends to make blueberries break during the harvesting process, thus causing additional costs, and manual harvesting is time-consuming and inefficient, hindering the cultivation and promotion of blueberries [1,2]. China possesses a vast territory, and manual harvesting is the main method for harvesting blueberries [3–9]. Due to the small size of the blueberry fruit and difficulties in harvesting, the harvesting work consumes a considerable number of manpower and material resources, which has become a bottleneck in the development of the blueberry industry chain.

Although the United States of America (USA) is the first country to carry out and engage in the research of blueberry mechanical harvesting, the operation mode in the USA does not match the planting agronomy in China. And the blueberry planting ridge

spacing and plant spacing in the USA are larger than those in China. For all these reasons, the harvester made in the USA cannot be directly used in blueberry harvesting operation in China.

At the same time, because the price of the blueberry harvester in the USA far exceeds the expected price of Chinese blueberry growers, coupled with the tariff barrier function, China has not yet introduced harvesters from the USA, and blueberry harvesting is still carried out manually. Therefore, it is necessary to study the blueberry harvesting mechanism according to the growth environment of blueberry plants and develop a domestic blueberry harvester to improve the mechanical harvesting of Chinese blueberries as soon as possible and broaden the development of the Chinese blueberry industry [10–12].

Studies in the related literature have shown that the "collision" in the harvesting operation is key to generating vibrational excitation and achieving fruit harvesting [13–20]. In the mechanical harvesting of blueberries, when the harvester finger rows and the blueberry plants collide with each other, the body of the harvester and the plants goes through a process of separation–collision–separation, forming a harvesting excitation force to make the plant vibrate and deform, and then the fruit harvesting inertia force is generated to separate the fruit from the branch.

Foreign researchers have carried out detailed research on the "collision" problem and achieved rich research results [21–26]. Among them, Hunt and Crossely have developed an L-N nonlinear spring damping model [23,24], which fitted the actual working conditions of blueberry harvesting and could be used to analyze the rigid–flexible coupling collision process of blueberry harvesting. Hossein et al. used L-N nonlinear spring damping model in their study of apple harvesting to achieve good application results [18].

To resolve the friction problem of the gap collision point, scholars at home and abroad first used the Coulomb friction model for analysis. The Coulomb model was a basic tribological model [26], which was able to describe the elastic collision between rigid bodies, but was not suitable for rigid–flexible coupling collision analysis. Bai's improved Coulomb friction model [26] was used in this paper and dynamic friction coefficients were introduced to consider the effect of static friction on the system. At the same time, the model could more accurately reflect the tangential contact characteristics, which was in line with the description of the tangential friction at the collision point of blueberry harvesting. Ebrahim et al. applied the improved Coulomb model to analyze the impact forces during peach transportation to provide a reference for engineering practice [19].

Little research has been performed in China on the interaction between harvesters and blueberry plants in terms of harvesting collision mechanisms [27,28]. Moreover, other project teams have not established flexible body models for blueberry plants or conducted related harvesting collision studies, except for our research group [29].

In summary, most of the literature has studied rigid body–rigid body collision, but there are few studies on rigid body–flexible body collision [30–37]. On the topic of fruit harvesting, especially regarding blueberry harvesting, there was even less literature on the collision between the rigid body of the harvester and the flexible body of the plant, which was the "core problem" in blueberry harvesting operations.

Based on the above analysis, this paper focuses on the collision process between the harvesting device of the harvester and the blueberry plant after manual pruning under the interaction of rigid–flexible coupling. The L-N nonlinear spring damping model and improved Coulomb friction model were used to establish a blueberry harvesting collision model, the harvesting collision force and tangential friction at the collision point were analyzed, and a theoretical study of the blueberry harvesting collision process was conducted. Based on the knowledge of multi-body dynamics, integrated finite element software and dynamics software, the harvesting collision force and fruit harvesting force generated by the interaction between the harvesting device and the blueberry plant were simulated and analyzed during the harvesting collision process. Furthermore, the influence of the working parameters of the harvester on the collision harvesting force was analyzed, and the influence of the working parameters on the harvesting efficiency and fruit harvesting

quality was experimented. All the above laid a theoretical foundation for the development of a blueberry harvester in China. In addition, the research results of this paper in the field of blueberry harvesting on the collision mechanism of rigid–flexible coupling can provide a theoretical reference for the harvesting of other similar berry shrubs.

## 2. Materials and Methods

### 2.1. Overview of the Research

In this paper, relevant research results at home and abroad were referred and the harvesting collision mechanism in the blueberry harvesting process was analyzed. The L-N nonlinear spring damping model and the improved Coulomb friction model were selected as the rigid–flexible coupling collision model for the interaction between the harvester and the blueberry plant. Thus, the collision theory equations related to this paper were obtained, as shown in Table 1.

**Table 1.** Theoretical equations related to blueberry harvesting collision.

| Sequence Number of Equations | Explanation of the Equations |
| :---: | :---: |
| 1 | Orthogonal decomposition and classification expressions for the collision force |
| 5 | Equation of the normal collision force at the collision point |
| 6 | Equation for the normal deformation of the blueberry branch at the collision point |
| 7 | Equation of the equivalent radius of curvature at the collision point of the blueberry branch |
| 8 | Equation of the equivalent elasticity modulus at the collision point of the blueberry branch |
| 9 | Equation for the equivalent damping coefficients of the normal collision force of the blueberry branch |
| 10 | Equation for the energy hysteresis damping factor in the collision process |
| 11 | Equation for tangential collision force at the collision point |
| 12 | Equation of friction coefficient for the tangential collision force at the collision point |
| 13 | Equation for the vector decomposition of the collision force of the blueberry branch |

A review of the literature revealed that MATLAB was a good scientific computing software and has been adopted by many scholars and has obtained good results, which was very suitable for this study. Therefore, this paper used MATLAB software to analyze the interaction collision process between the harvester and the blueberry branch. Since the object of this paper was the collision force, where the tangential collision force was difficult to be measured by the field test, ADAMS software was used to analyze the blueberry harvesting collision process in order to explore the correctness of this study. ADAMS software was used because ADAMS is a professional multi-body dynamics simulation software, mainly used to analyze the interaction between objects, which was consistent with the research content of this paper. The theoretical basis of the "collision" tool supported by ADAMS software was in line with the theory of this study, and the analysis results possess some theoretical reference value for this study. It was undeniable that the simulation analysis results of the above two software might have some differences with the actual occurrence of the collision force, which was the limitation of this study.

The flow chart of this study is shown in Figure 1. Firstly, the structure and working principle of blueberry harvester were studied, the harvesting collision process was analyzed, and the rigid–flexible coupling collision model of the blueberry branch was established in MATLAB and ADAMS, respectively.

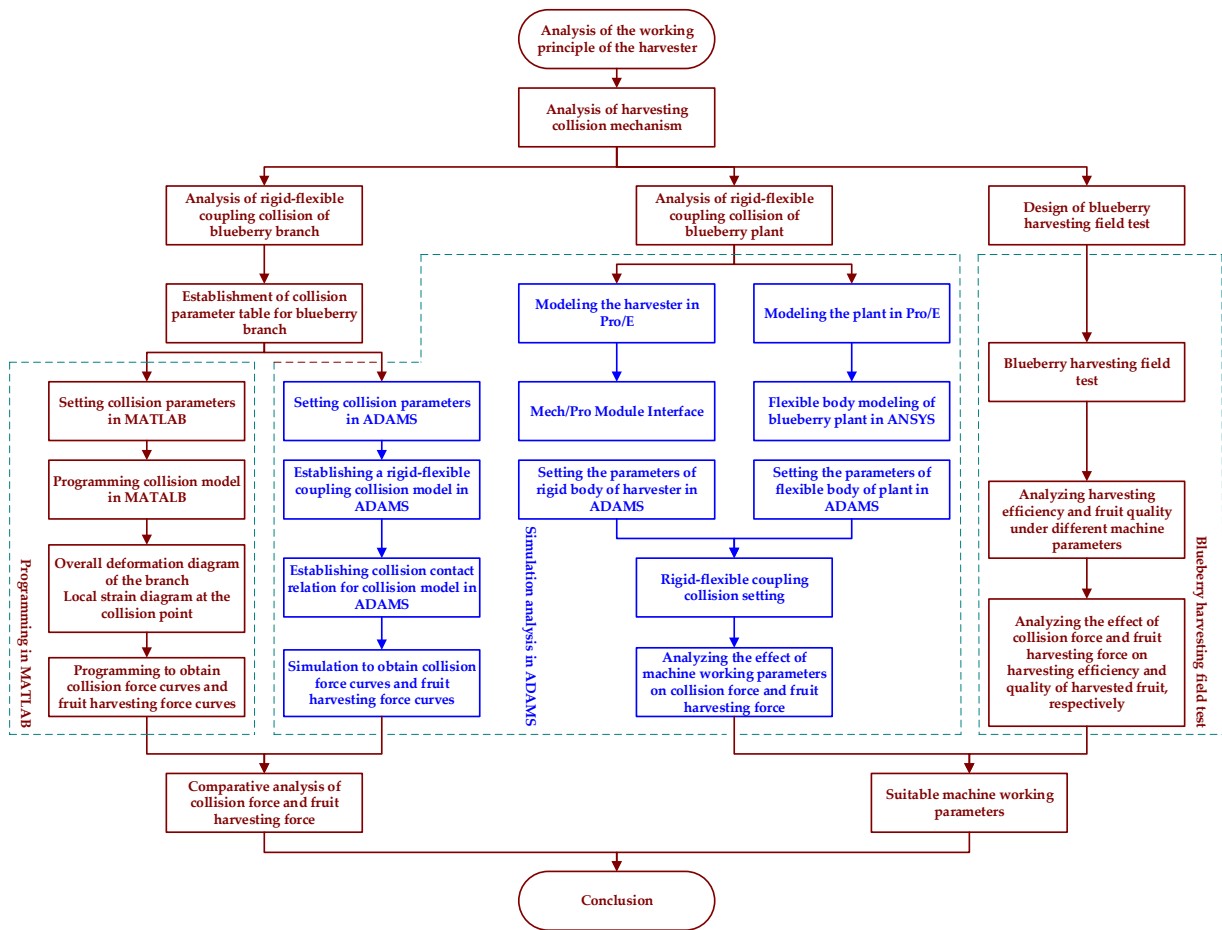

**Figure 1.** Flowchart of the research.

Secondly, the collision parameter was set in the above two software with reference to the collision parameter table (in Section 2.4.1). The collision model was programmed and analyzed in MATLAB; the collision model was simulated in ADAMS; and the collision force curves of branch and fruit harvesting force curves obtained by the two software were studied in comparison.

Thirdly, the flexible body model of blueberry plant and the rigid body model of harvester were established by integrating Pro/E, ANSYS, and ADAMS software. Then, the rigid–flexible coupling connection settings of the model were made with reference to the collision parameter table. The effects of machine working parameters on the collision force and fruit harvesting force were analyzed in ADAMS.

Finally, under different machine parameters, the impact of collision force and fruit harvesting force on harvesting efficiency and the quality of harvested fruit were analyzed with a blueberry harvesting field test. Then, the machine parameters of the harvester were comprehensively obtained to meet the harvesting conditions.

## 2.2. Rigid–Flexible Coupling Collision Harvesting Mechanism

### 2.2.1. Structure and Working Principle of the Blueberry Harvester

The structure and working diagram of the harvester is shown in Figure 2, mainly composed of the following parts: a tractor, a traction device, a gathering device, finger rows, a hydraulic system, a gantry frame, running system, a fruit harvesting device, a horizontal conveyor belt, and an inclined conveyor belt. The length of the machine was 3.0 m, the width was 1.9 m, and the height was 2.0 m. The machine was driven by a hydraulic system and the output rotational velocity of the harvesting system ranged from 50 r/min to 250 r/min.

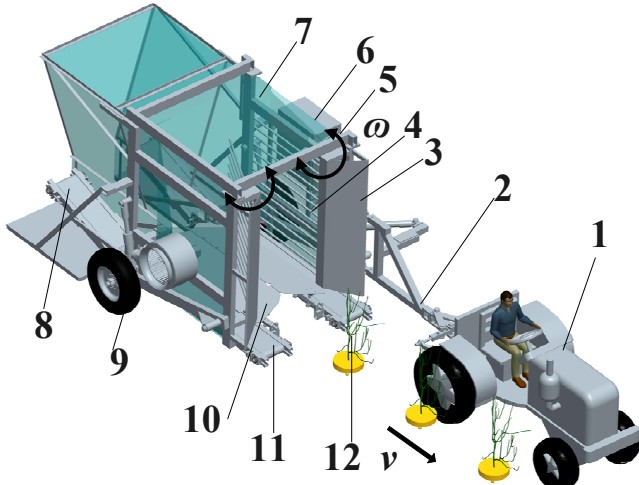

**Figure 2.** Structure and working diagram of the blueberry harvester. 1. tractor; 2. traction device; 3. gathering device; 4. finger rows; 5. hydraulic motor; 6. hydraulic system; 7. gantry frame; 8. inclined conveyor; 9. running system; 10. fruit catching device; 11. horizontal conveyor; 12. blueberry plants. Note: $v$ is the machine velocity of the harvester, (m/min); $\omega$ is the angular velocity of the swing of the finger rows, (r/min).

The figure shows a working schematic diagram of the harvester, and the details of the working principle are as follows. During the blueberry harvesting process, the harvester rides the ridge for operation, the body of which moves forward at a certain velocity $v$; the blueberry plant enters the harvester gantry frame through the gathering device; the hydraulic system drives the hydraulic motor of the harvesting device to rotate, thus driving the finger rows on the left and right sides to beat the blueberry plant passing through it at a certain rotational velocity $\omega$; the finger rows on both sides collide with the blueberry plant to form a vibration and vibrate off the blueberry fruit; and the blueberry fruit being vibrated off is transported to the designated position through the fruit catching device, the horizontal conveyor, and the inclined conveyor, finally achieving the mechanical harvesting of blueberry.

2.2.2. Mechanistic Analysis of Rigid–Flexible Coupling Collision Harvesting

In the blueberry harvesting operation, the finger rows on both sides of the harvester form parallel force application units, which interact with the flexible body of the plant at all levels of branching to form a multi-point stimulated harvesting collision force vector $\overrightarrow{F}_i(t)$ at the point of contact, as shown in Figure 3.

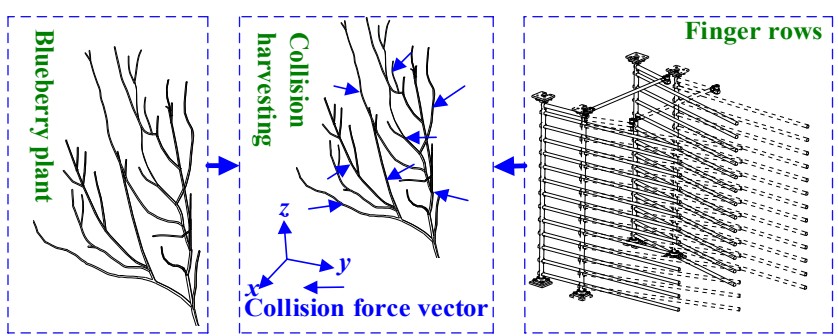

**Figure 3.** Diagram of multipoint collision of blueberry harvesting.

A simplified force analysis of a single finger row and a single blueberry branch at the moment of collision is shown in Figure 4. As the finger row is machined from metal pipe,

its stiffness value is much greater than that of the blueberry branch, which can be assumed to be a rigid body without deformation; the blueberry branch with good deflection and toughness is simplified to a flexible body due to greater bending deformation when the two interact.

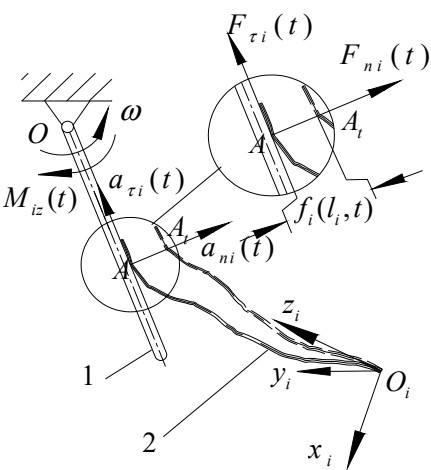

**Figure 4.** Collision analysis of blueberry harvesting. 1. Finger rows; 2. blueberry branch. Note: $A$ is the collision point when the blueberry plant is not deflected; $A_t$ is the collision point when the blueberry plant is deflected; $f_i(l_i, t)$ is the deflection of the blueberry plant at the collision point, (m); $F_{ni}(t)$ is the normal collision force at the point of impact of the harvester's finger rows on the blueberry branch (N); $F_{\tau i}(t)$ is the tangential collision force at the point of impact of the harvester's finger rows on the blueberry branch (N); $M_{iz}(t)$ is the driving torque of the finger rows (N·m); $\alpha_{ni}$ is the normal collision acceleration of the branch (m/s$^2$); $\alpha_{\tau i}$ is the tangential collision acceleration of the branch (m/s$^2$); $x_i y_i z_i$ is the i-grade branching coordinate system.

At the moment of contact and collision between the harvesting device and the blueberry branch, the rigid finger rows of the harvester interact with the flexible body of the blueberry plant at a certain velocity $\omega$ and a certain oscillation frequency, which creates a transient collision harvesting force vector $\overrightarrow{F}_i(t)$ at point $A$ of the blueberry branch, as shown in Figure 4a. To facilitate the analysis, the collision harvesting force vector $\overrightarrow{F}_i(t)$ is orthogonally decomposed at the collision point to obtain the normal collision force $F_{ni}(t)$ and the slip friction force $F_{\tau i}(t)$, and $\overrightarrow{F}_i(t)$ can be expressed as:

$$\overrightarrow{F}_i(t) = F_{ni}(t)\overrightarrow{i} + F_{\tau i}(t)\overrightarrow{j} \tag{1}$$

where $\overrightarrow{F}_i(t)$ is the collision force vector at the point of impact of the harvester's finger rows on the blueberry branch (N).

The normal collision force $F_{ni}(t)$ causes the blueberry branch to deform in deflection $f_i(l_i, t)$, the deformation of which gradually changes from the root point $O_i$ of the branch to the end. When the collision point $A$ moves to point $A'$ at time $t$, the collision produces a harvesting force $F_{maxi}^{f}(t)$ on the fruit on the branch. The deflection deformation at i-grade branch is obtained from mechanical vibration science and can be expressed as [27,29]:

$$f_i(l_i, t) = A_i(l_i) \cdot e^{-\xi \omega_i t} \cdot \sin(\omega_i \cdot t + \phi_i) \tag{2}$$

where $f_i(t)$ is the deflection deformation of the i-grade branch (m); $A_i(l_i)$ is the i-grade branching mode shape function, which is a function of the growth position and normal collision force of blueberry fruit (m); $\xi$ is the damping ratio of the blueberry branch due to structural damping; $\phi_i$ is the initial phase angle of the i-grade branch (rad); $\omega_i$ is the blueberry branch vibration angular frequency (rad/s); $e$ is the natural logarithm.

From Equation (2), the maximum blueberry fruit harvesting force $F_{maxi}^f$, when the flexible body of the blueberry plant is impacted by the vibration of a finger rows of rigid bodies, is:

$$F_{maxi}^f = m_0 A_i(l_i)\omega_i^2 \sin\phi_i \cdot \left(1 - \xi^2\right) \tag{3}$$

where $F_{\max i}^f$ is the maximum blueberry fruit harvesting force ($F_{maxi}^f = max\left\{F_i^f(t)\right\}$ $F_i^f(t)$ is the harvesting force), when the i-grade branch is impacted by the vibration of finger rows (N); $m_0$ is the mass of the blueberry fruit.

From the above conditions, it can be obtained that when the harvester is in operation, the conditions for vibratory harvesting of ripe blueberry fruit is as follows:

$$F_2 < F_{maxi}^f < F_1 \tag{4}$$

where $F_1$ is the bonding force between the branch and the raw fruit, $F_1 = 1.0 - 3.6$ (N) [5,7]; $F_2$ is the minimum bonding force between the branch and the ripe fruit, $F_2 = 0.26 - 0.3$ (N) [5,7].

If $F_{maxi}^f$ is greater than the bonding force between the branch and the raw fruit, then the harvested blueberry fruit will be interspersed with raw fruit. If $F_{maxi}^f(t)$ is less than the bonding force between the blueberry fruit and the branch, then, some ripe fruit that is not shaken off will remain on the branch after the harvester has acted, which is not vibrated.

### 2.3. Modelling of the Rigid–Flexible Coupling Collision Force

### 2.3.1. Modelling of the Normal Collision Force

An L-N nonlinear spring damping model was applied in this paper to analyze the blueberry harvesting collision force. As shown in Figure 5, at the moment of collision between the harvester finger rows and the blueberry plant, the blueberry plant undergoes collisional deformation $\delta_{in}$, generating a normal collision force $F_{ni}$, which can be expressed as:

$$F_{ni}(t) = k_i \delta_{ni}^q(t) + c_i \dot{\delta}_{ni}(t) \tag{5}$$

where $\delta_{ni}(t)$ is the amount of deformation in the direction normal to the collision of the blueberry plant (m); $q$ is the power exponent of elastic deformation; $k_i$ is the equivalent stiffness coefficient (N/m); $c_i$ is the equivalent damping coefficient (N.s/m).

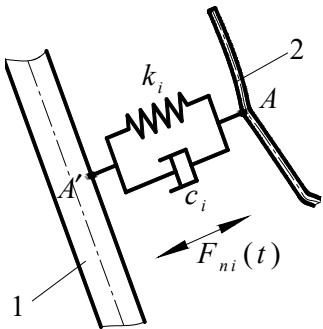

(**a**) Normal collision model for the branch

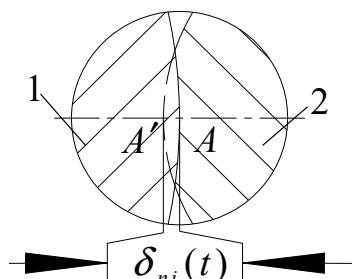

(**b**) Normal collision deformation diagram of the branch

**Figure 5.** Nonlinear elastic damping model for the normal collision force in blueberry harvesting. Note: $\delta_{ni}(t)$ is the normal deformation of the blueberry branch at the point of collision (μm). 1. Finger rows 2. Blueberry branch.

Based on the L-N spring damping model, the following relationship exists between the collision force $F_{ni}$ and the collision deformation $\delta_{ni}$ in Equation (5):

$$\delta_{ni}(t) = \left( \frac{9F_{ni}^2(t)}{16ER_i^2} \right)^{\frac{1}{3}} \tag{6}$$

where $R_i$ is the equivalent radius of curvature at the collision point (m); $E$ is the equivalent elasticity modulus at the collision point (N/m).

The equivalent radius of curvature $R_i$ at collision point for Equation (6) is expressed as:

$$R_i = \frac{R_{1i}R_{2i}}{R_{1i} + R_{2i}} \tag{7}$$

where $R_{1i}$ is the radius of curvature of the finger rows at the collision point (m); $R_{2i}$ is the radius of curvature of the blueberry branch at the point of collision (m).

The equivalent elasticity modulus $E$ at the point of impact of Equation (6) can be expressed as:

$$E = \frac{E_1 E_2}{\left(1 - \mu_2^2\right)E_1 + \left(1 - \mu_1^2\right)E_2} \tag{8}$$

where $E_1$ is the elasticity modulus of the finger rows (N/m); $E_2$ is the elasticity modulus of the blueberry branch (N/m); $\mu_1$ is the Poisson's ratio of the finger rows; $\mu_2$ is the Poisson's ratio of the blueberry branch.

Combining Equations (5) to (8), it can be yielded that the power exponent of elastic deformation $q = \frac{3}{2}$ and the equivalent stiffness coefficient $k_i$.

Since the nonlinear damping coefficients in the L-N spring damping model are continuous functions, however, the equivalent damping coefficients are stepwise as the blueberry harvesting goes through the separation–collision–separation process, the STEP function is introduced to make corrections as follows [37]:

$$c_i = \begin{cases} 0 & \delta_{ni} \leq 0 \\ \frac{3\delta_{ni}^2}{d_{max}} - \frac{2\delta_{ni}^3}{d_{max}^2} & 0 \leq \delta_{ni} \leq d_{max} \\ \zeta_i \cdot \delta_{ni}^q & \delta_{ni} \geq d_{max} \end{cases} \tag{9}$$

where $d_{max}$ is the maximum penetration depth at the collision point, $0 \leq d_{max} \leq 5 \times 10^{-5}$ [37,38] (m); $\zeta_i$ is the energy hysteresis damping factor for the collision process.

The energy hysteresis damping factor in the collision process $\zeta_i$ is expressed as:

$$\zeta_i = \frac{3k_i\left(1 - e^2\right)}{4\left|v_{1i}^- - v_{2i}^-\right|} \tag{10}$$

where $v_{1i}^-$ is the finger velocity before collision (m/s); $v_{2i}^-$ is the blueberry branch velocity before the collision (m/s); $e$ is the collision recovery coefficient, which is related to the material of the collision object. When the collision is fully elastic and there is no energy loss, $e = 1$; when colliding at low velocity and with energy consumption, $e \to 1$; in the event of a fully plastic collision, $e = 0$; in this paper, the elastic collision of the rows and the blueberry plant is a low-velocity collision, and the energy is not lost, so $e = 1$ is taken.

2.3.2. Modelling of the Tangential Collision Force

In the improved Coulomb friction model, the collision point slip friction $F_{iz}$ can be expressed as:

$$F_{\tau i}(t) = F_{ni}(t) \cdot \mu\left(\dot{\delta}_\tau\right) \tag{11}$$

where $\dot{\delta}_\tau$ is the relative sliding velocity of the collision point in the tangent direction (m/s); $\mu\left(\dot{\delta}_\tau\right)$ is the coefficient of friction function of the collision point, which is expressed as:

$$\mu\left(\dot{\delta}_\tau\right) = \begin{cases} -\mu_d, \dot{\delta}_\tau \in \left(\dot{\delta}_d, +\infty\right) \\ -\mu_d - (\mu_s - \mu_d)\left(\frac{\dot{\delta}_\tau - \dot{\delta}_d}{\dot{\delta}_s - \dot{\delta}_d}\right)^2 \cdot \left(3 - 2\frac{\dot{\delta}_\tau - \dot{\delta}_d}{\dot{\delta}_s - \dot{\delta}_d}\right), \dot{\delta}_\tau \in \left[\dot{\delta}_s, \dot{\delta}_d\right] \\ \mu_s - 2\mu_s\left(\frac{\dot{\delta}_\tau + \dot{\delta}_s}{2\dot{\delta}_s}\right)^2 \cdot \left(3 - 2\frac{\dot{\delta}_\tau + \dot{\delta}_s}{2\dot{\delta}_s}\right), \dot{\delta}_\tau \in \left[-\dot{\delta}_s, \dot{\delta}_s\right] \\ \mu_d + (\mu_s - \mu_d)\left(\frac{\dot{\delta}_\tau - \dot{\delta}_d}{\dot{\delta}_s - \dot{\delta}_d}\right)^2 \cdot \left(3 - 2\frac{\dot{\delta}_\tau - \dot{\delta}_d}{\dot{\delta}_s - \dot{\delta}_d}\right), \dot{\delta}_\tau \in \left[-\dot{\delta}_d, -\dot{\delta}_s\right] \\ \mu_d, \dot{\delta}_\tau \in \left(-\infty, -\dot{\delta}_d\right) \end{cases} \quad (12)$$

where $\mu_s$ is the coefficient of static friction at the collision point; $\mu_d$ is the coefficient of the sliding friction at the collision point; $\dot{\delta}_s$ is the critical velocity at which static friction turns into sliding friction (m/s); $\dot{\delta}_d$ is the critical velocity at which kinematic friction turns into the sliding friction (m/s). The relation diagram of relative sliding velocity and friction coefficient is shown in Figure 6.

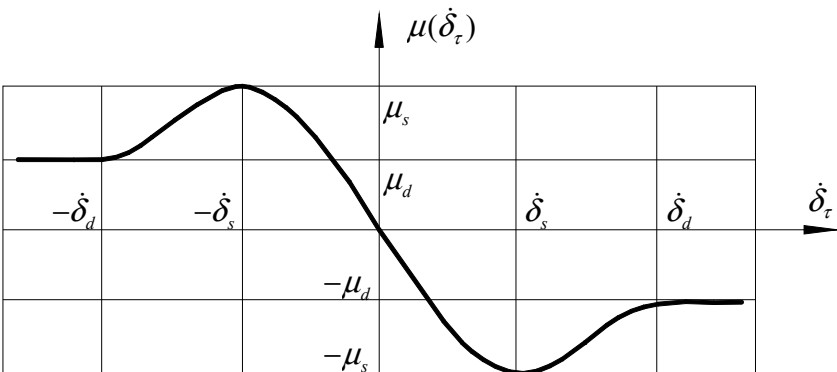

**Figure 6.** Relation diagram of relative sliding velocity and friction coefficients.

### 2.3.3. Vector Analysis of the Collision Force

The harvesting collision force obtained by the above solution refers to the harvesting collision force under the row coordinate system. Since the research object of the machine harvesting operation is the blueberry plant, the harvesting collision force under the coordinate system of the finger rows should be converted into the harvesting collision force under the blueberry plant coordinate system. A schematic diagram of the orthogonal decomposition of normal collision forces and tangential slip friction forces is shown in Figure 7.

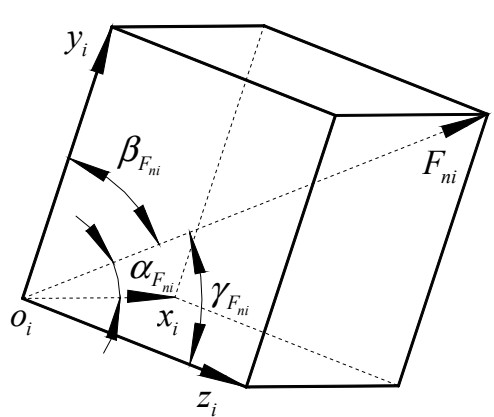

(**a**) The orthogonal decomposition of the normal collision force

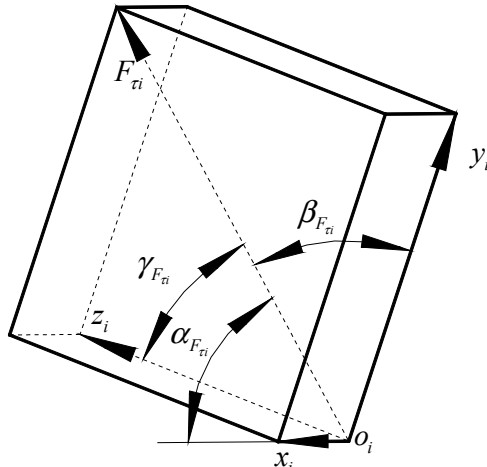

(**b**) The orthogonal decomposition of the tangential collision force

**Figure 7.** Schematic diagram of the orthogonal decomposition of the fruit harvesting collision force under the i-grade branch. Note: $\alpha_{F_{ni}}$ is the angle between the normal collision force $F_{ni}$ and the $x_i$ axis of the i-grade branching coordinate system $x_i y_i z_i$ (rad); $\beta_{F_{ni}}$ is the angle between the normal collision force $F_{ni}$ and the $y_i$ axis of the i-grade branching coordinate system $x_i y_i z_i$ (rad); $\gamma_{F_{ni}}$ is the angle between the normal collision force $F_{ni}$ and the $z_i$ axis of the i-grade branching coordinate system $x_i y_i z_i$ (rad); $\alpha_{F_{\tau i}}$ is the angle between the tangential collision force $F_{ni}$ and the $x_i$ axis of the i-grade branching coordinate system $x_i y_i z_i$ (rad); $\beta_{F_{\tau i}}$ is the angle between the tangential collision force $F_{ni}$ and the $y_i$ axis of the i-grade branching coordinate system $x_i y_i z_i$ (rad); $\gamma_{F_{\tau i}}$ is the angle between the tangential collision force $F_{ni}$ and the $z_i$ axis of the i-grade branching coordinate system $x_i y_i z_i$ (rad).

From Figure 7, the expression of the harvesting collision force under each coordinate system of the i-grade branch of the blueberry plant can be obtained:

$$
\begin{aligned}
\vec{F}_i(t) = & \left[ F_{ni}(t) \cdot \cos\alpha_{F_{ni}} + F_{\tau i}(t) \cdot \cos\alpha_{F_{\tau i}} \right] \vec{i} \\
& + \left[ F_{ni}(t) \cdot \cos\beta_{F_{ni}} + F_{\tau i}(t) \cdot \cos\beta_{F_{\tau i}} \right] \vec{j} \\
& + \left[ F_{ni}(t) \cdot \cos\gamma_{F_{ni}} + F_{\tau i}(t) \cdot \cos\gamma_{F_{\tau i}} \right] \vec{k}
\end{aligned}
\tag{13}
$$

*2.4. Simulation Setup of Harvesting Collision and Harvesting Test Design of Blueberry*

2.4.1. Simulation Setup for Single-Point Collision of Blueberry Branch under MATLAB

According to [2,37–43], the dimensional parameters, mechanical parameters, and kinematic parameter of the blueberry plant and the finger rows of harvester were set separately. The dimensional, mechanical, and kinematic parameters of the blueberry branch and the finger rows of the harvester were set according to the serial numbers 1 to 12 and 15 to 24 in Table 2, and a numerical simulation of the single-point collision of the blueberry branch was set in MATLAB. Programming with MATLAB allowed the single-point collision process of the blueberry branch to be measured.

**Table 2.** Summary table of parameter descriptions.

| Sequence Number | Parameter | Parameter Name | Unit | Parameter Name | Literature Resources | Notes |
|---|---|---|---|---|---|---|
| 1 | $R_{2i}$ | Radius of curvature of the branch | (mm) | 5 | | Measured in the blueberry field |
| 2 | $l_i$ | Branch length | (mm) | 400 | | Measured in the blueberry field |
| 3 | $E_2$ | Elasticity modulus of branch | (MPa) | 690 | [40] | |
| 4 | $\mu_2$ | Poisson's ratio of branch | | 0.3 | [2] | |
| 5 | $v_{2i}^-$ | Velocity of branch before collision | (m/s) | 0 | | |
| 6 | $\rho_2$ | Density of branch | (kg/m³) | $0.9 \times 10^3$ | | Measured in the test |
| 7 | $\xi$ | Damping ratio of branch | | 0.1 | [41] | |
| 8 | $l_{point}$ | Location of branch collision point | (mm) | 200 | | Midpoint of the branch |
| 9 | $l_{fruit}$ | Position of fruit growth on the branch | (mm) | 400 | | End of the branch |
| 10 | $m_0$ | Fruit mass | (g) | 2 | | Measured in the test |
| 11 | $R_{1i}$ | Radius of curvature of the finger rows | (mm) | 15 | | Mechanical parameters of the harvester |
| 12 | $l_1$ | Length of the figure rows | (mm) | 600 | | Mechanical parameters of the harvester |
| 13 | $\Delta d$ | Spacing of finger rows | (mm) | 200 | | Mechanical parameters of the harvester |
| 14 | n | Number of single-sided finger rows | (PCS) | 15 | | Mechanical parameters of the harvester |
| 15 | $\rho_1$ | Density of the finger rows | Kg/m³ | $1.15 \times 10^3$ | | Measured in the test |
| 16 | $E_1$ | Elasticity modulus of the figure rows | (MPa) | $30 \times 10^3$ | [40] | |
| 17 | $\mu_1$ | Poisson's ratio of the figure rows | | 0.25 | [42] | |
| 18 | $v_{1i}^-$ | Velocity of the figure rows before collision | (m/s) | 6 | | |
| 19 | $\mu_s$ | Coefficient of static friction | | 0.1 | [2] | |
| 20 | $\mu_d$ | Coefficient of sliding friction | | 0.05 | [41] | |

**Table 2.** *Cont.*

| Sequence Number | Parameter | Parameter Name | Unit | Parameter Name | Literature Resources | Notes |
|---|---|---|---|---|---|---|
| 21 | $\dot{\delta}_s$ | Critical velocity at which static friction turns into sliding friction | (m/s) | 0.5 | [43] | |
| 22 | $\dot{\delta}_d$ | Critical velocity at which kinematic friction turns into sliding friction | (m/s) | 1 | [43] | |
| 23 | $d_{max}$ | Maximum penetration depth at the collision point | (mm) | 0.05 | [38,39] | |
| 24 | $e$ | Collision recovery coefficient | | 1 | [38,39] | |
| 25 | $q$ | Power exponent | | 1.5 | | Calculated from Equations (5)–(8) |

2.4.2. Simulation Setup for Single-Point Collision of Blueberry Branch under ADAMS

In the ADAMS environment, a flexible body model of the blueberry branch was created by stretching method; one end of the flexible body was set as a fixed end (set as the root of the blueberry branch) and one end was set as a free end (set as the end of blueberry branch). The unit type of the flexible body was Solid Hex, and the cross section of the branch was set as Elliptical. In order to accurately compare and analyze the simulation data, the parameters of serial numbers 1 to 12 and 15 to 25 in Table 2 were selected to set the dimensional, mechanical, and kinematic parameters of the blueberry branch and the finger rows of harvester, respectively.

The connection between the two was set to be a rigid–flexible collision connection, and the impact function method was chosen to calculate the collision force of the model according to Equations (6) to (10). The force index $q$, damping ratio $\xi$, and penetration depth $d_{max}$ were set to the corresponding parameters and specific values in Table 2. The serial numbers of the parameters were 25, 7, and 23, respectively.

The Coulomb model was used to calculate the friction force of the collision model according to Equations (11) and (12). The coefficient of static friction $\mu_s$ at the collision point, the coefficient of kinematic friction $\mu_d$, the critical velocity at which static friction turns into sliding friction $\dot{\delta}_s$, and the critical velocity at which kinematic friction turns into sliding friction $\dot{\delta}_d$ were set to the corresponding parameters and specific values in Table 2. The serial numbers of the parameters were 19–22, respectively. The RKF45 (Runge–Kutta method) was used to iteratively calculate the collision process. The collision time was set to 1 s, the integration step was set to 0.01 s; at the initial moment of the collision, the distance between the finger rows and the branch was set to 200 mm; the collision point was the midpoint of the branch; the branch was stationary; the collision velocity of the finger rows was set to 6 m/s (the serial numbers of the parameters were 5, 8, and 18); the fruit growth position was set to the end of the branch; and the mass of the blueberry fruit was set to the corresponding parameter and specific values in Table 2 (the serial numbers of the parameters were 9 and 10). Based on the above setting, the ADAMS software was used to analyze the single-point collision process of the blueberry branch.

### 2.4.3. Simulation Setup for Multi-Point Collision of Blueberry Plant

The growth structure of blueberry plant after manual pruning was analyzed, based on the modeling principle of L-system, and Pro/E software was used to build the growth model of blueberry plant, which was imported into ANSYS software. Secondly, the finite element model of the blueberry plant was obtained by meshing, which was saved as a $*.mnf$ modal neutral file and exported to ADAMS, and the root of the plant was set to be connected to the earth. The plant roots were set to be connected to the earth and the connection type was fixed joint. The density of the plant ρ, the elastic modulus $E_2$, the damping ratio $\xi$, and Poisson's ratio $\mu_2$ were set to the corresponding parameters and specific values in Table 2 (the serial numbers of the parameters were 6, 3, 7, and 4), respectively. Then, the flexible body model of the blueberry plant was obtained, as shown in Figure 8a,c.

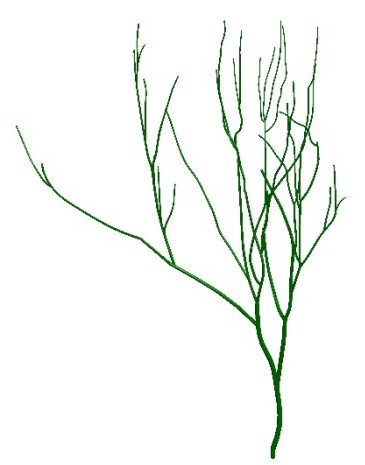

(**a**) Pro/E model of a blueberry plant

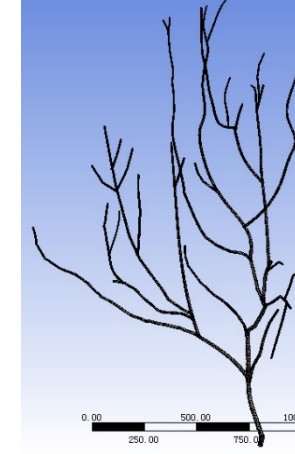

(**b**) ANSYS finite element model of a blueberry plant

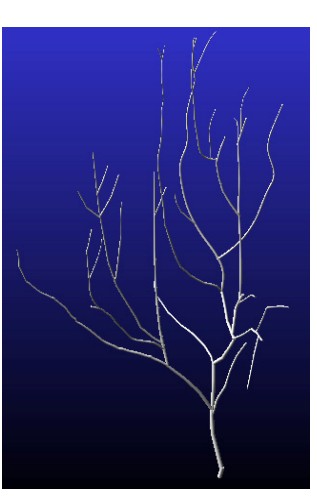

(**c**) ADAMS model of a blueberry plant

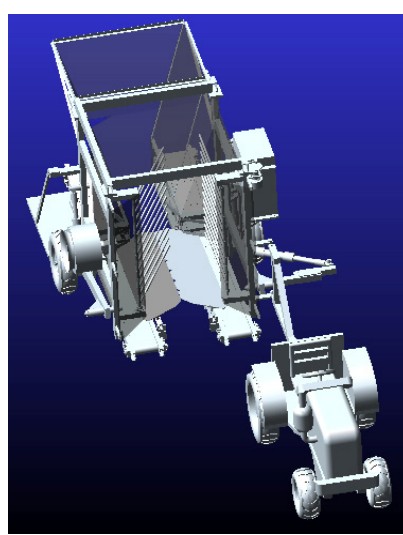

(**d**) Rigid body modelling of a harvester in ADAMS

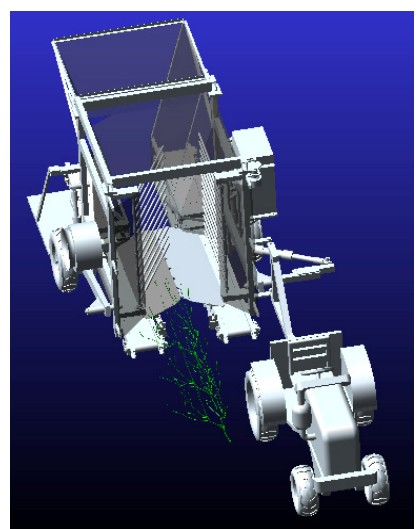

(**e**) A rigid–flexible coupling dynamics model of a harvester and a blueberry plant

**Figure 8.** Modelling the rigid–flexible coupling collision harvesting process.

Based on the structural parameters of the harvester, Pro/E software was used to model the harvester, and the model created was imported into ADAMS software through the Mech/Pro interface. Then, the materials of the components of the harvester were set

separately, and the connections of the moving parts were set. Therefore, the rigid body model of the harvester was obtained, as shown in Figure 8d.

In the ADAMS environment, the rigid body model of the harvester and the flexible body model of the blueberry plant were combined, and the connection between the two was set to a rigid–flexible coupling collision connection, with the collision parameters set in line with the collision settings of ADAMS software in Section 2.4.2 (the sequence numbers of the parameters were 10–17 and 19–25). The relative positions of the harvester and the blueberry plant were set to maintain a distance of 400 mm at the initial moment of the simulation, and blueberry plants remained stationary. The harvester set a certain machine velocity and the driving elements of the harvesting system set a certain rotational velocity. Then, a rigid–flexible coupled dynamics model of the harvester and the blueberry plant was established, as shown in Figure 8e.

Observation of the blueberry harvesting process in the plantation showed that the collision points between the harvester finger rows and the blueberry branch were mostly concentrated in the middle of the plant and the middle of the branch; the blueberry fruit clusters were mostly distributed in the middle of the plant and the end of the branch.

Based on this, the following settings were made for the simulation test environment: the middle part of the blueberry branch growing in the middle of the plant was set as the collision measurement point for the interaction between the harvester finger rows and the blueberry plant; the end position of the branch growing in the middle of the blueberry plant was set as the fruit harvesting force measurement point; and the mass of the blueberry fruit at the measurement point was set to 2 g (this is consistent with the values in Table 2). Changes in the collision force and fruit harvesting force under different conditions were simulated and analyzed.

2.4.4. Design of Blueberry Harvesting Field Test

Since the collision harvesting force formed by the interaction between the finger rows and the blueberry plant was a transient excitation force of multi-point action, which was difficult to be accurately measured, in the testing, the amount of the raw fruit harvested by the machine, the amount of damaged fruit, and the harvesting efficiency of the fruit were used as evaluation criteria to evaluate the collision force and the harvesting power of the fruit. The Wulongbei Blueberry Plantation in Zhen'an District, Dandong City, Liaoning Province, China, was used as the testing site. The testing time was July 2019, and the machine used was a self-propelled blueberry harvesting machine (as shown in Figure 9).

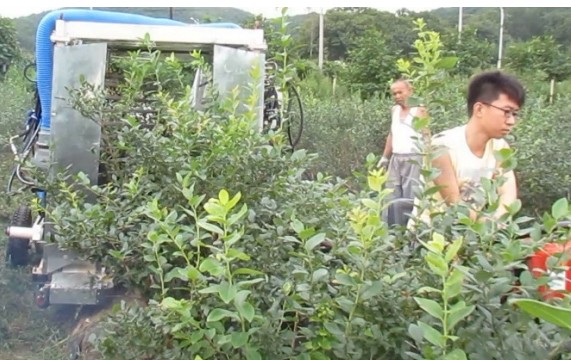

**Figure 9.** Blueberry harvesting field test.

The Huayuan 350 tractor (made in Shandong Province, China) was used for harvesting. The rated power of the tractor was 22 kW and its power was sufficient to drive the harvester for uniform machine velocity. The tractor had four speed gears, as well as a high gear and low-speed gear switching function. The harvesting system drive element was the 1QJM001-0.10 type hydraulic motor, and the rated torque of hydraulic motor was 154 N·m and the output rotational velocity ranged from 10 r/min to 500 r/min. The rotational

velocity unit of the hydraulic motor was the number of revolutions per minute, abbreviated as "r/min", similarly hereinafter. In the field test, the tractor drove forward in a low gear; the driving velocity is shown in Table 3.

**Table 3.** Table of working parameters of the harvester.

| Group Number | 1 | 2 | 3 | 4 | 5 | 6 |
|---|---|---|---|---|---|---|
| Rotational velocity (r/min) | 70 | 130 | 190 | 28 | 45 | 68 |
| Machine velocity (m/min) | 28 | 28 | 28 | 120 | 120 | 120 |

The test method was a single-factor method, which meant that the rotational velocity of the hydraulic motor of the harvesting system and the machine velocity of the harvester were used to obtain the amount of raw fruit harvested by the machine and the amount of the damaged fruit, calculate the fruit harvesting efficiency, and analyze the influence of the collision harvesting force and the fruit harvesting force on the quality of the fruit harvested and the harvesting efficiency of the machine in combination with the rigid–flexible coupling collision mechanism when the machine velocity of the harvester and the rotational velocity of hydraulic motor of drive device were changed (as shown in Table 3).

In order to analyze and compare the test data, the same land, the same species (Bluecrop), and the same age of blueberry plants were selected as the test subjects. In the blueberry harvesting operation, the machine velocity of the harvester and the output rotational velocity of the harvesting device were kept constant, and each harvesting operation time was set to last 30 s each; thus, a blueberry harvesting field test was obtained.

As the harvesting field test was a randomized test and the test conditions were the same, a statistical analysis of the completely randomized test method was adopted. Five tests (each with a harvesting time of 30 s) were conducted and the test data were counted while keeping the machine parameters constant. The data from five tests were calculated and averaged. The mean value was taken as the field test results under the condition of same machine parameters, resulting in the data in Table 3.

In order to analyze and evaluate the working performance of the harvester, it was necessary to calculate the harvesting efficiency of the machine, the harvesting rate of raw fruit, and the damage rate of harvested fruit. The various indicators of the test are defined below [28].

The harvesting efficiency: the mass of fruit harvested by the harvester in a unit of time, unit: kg/min. The harvesting rate of raw fruit: the amount (mass) of raw fruit harvested by the machine as a percentage of the total amount (mass) of fruit harvested, unit: %. The damage rate of harvested fruit: the amount (mass) of harvested damaged fruit by the machine as a percentage of the total amount (mass) of fruit harvested, unit: %.

The harvesting field test procedure was as follows. Prior to the harvesting test, the harvester first needed to be positioned so that the blueberry plants were in the middle of the finger rows and the working parameters of the harvester were set to the specified values (as shown in Table 3). In the mechanical fruit harvesting operation, the harvester was operated continuously and the working parameters were kept constant. After the harvesting test, the harvested blueberry fruits were statistically analyzed to obtain the test results of each blueberry harvesting operation, while the working parameters of the machine were readjusted according to the values in Table 3 to prepare for the next harvesting field test.

## 3. Results and Discussion

### 3.1. Simulation Analysis of Single-Point Collision of Blueberry Branch

3.1.1. Numerical Simulation by MATLAB to Analyze Single-Point Collision

According to the setting parameters by MATLAB in Section 2.4.1, combined with Equations from (5) to (8), MATLAB calculation programming was used to obtain the deformation surface of the branch at the collision point (Figure 10), the branch vibration response surface (Figure 11), the branch collision force curves (Figure 12), and the blueberry fruit harvesting force curves (Figure 13), all of which are shown in Figures 9–13.

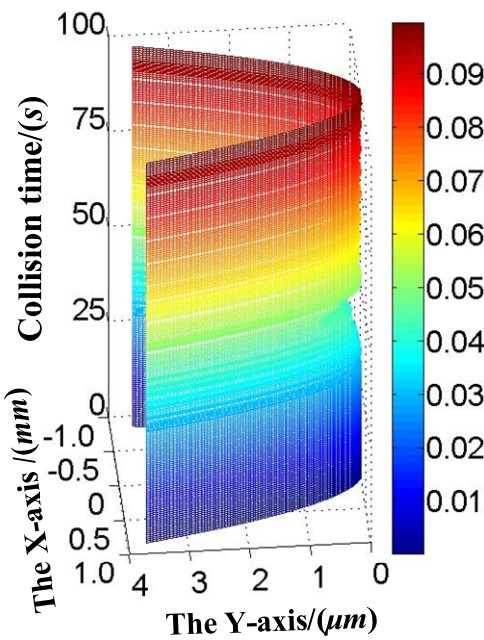

**Figure 10.** Strain diagram at the single collision point.

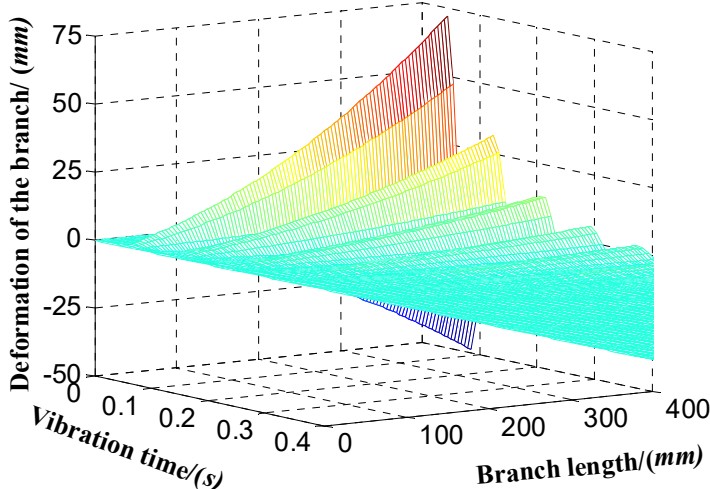

**Figure 11.** Deformation surface for the blueberry branch.

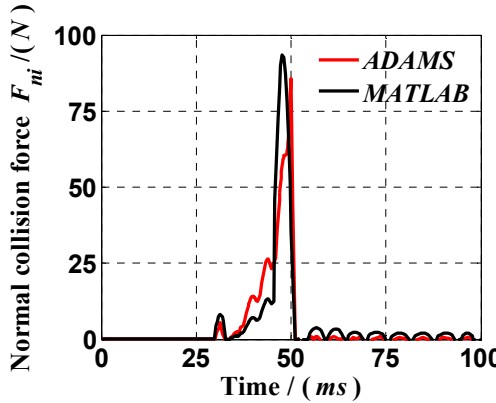

(**a**) Collision force curves for the normal force

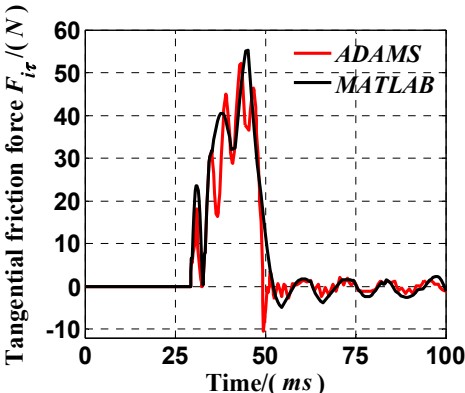

(**b**) Collision force curves for the tangential friction force

**Figure 12.** Collision force curves of the blueberry branch.

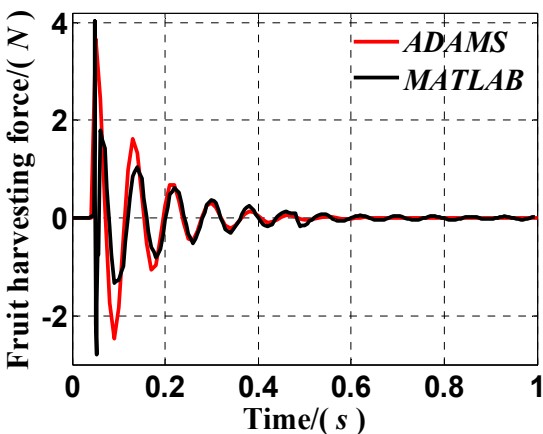

**Figure 13.** Fruit harvesting force curves of the blueberry branch.

As shown in Figure 10, the deformation surface of the branch at the collision point was analyzed. At 30 ms, the finger rows collided with the blueberry branch; as the collision process continued, the collision deformation began to increase; at 45 ms, the collision deformation reached its maximum, the value of which was about 0.5 μm; and then the collision deformation decreased, which tended to collide with deformation 0 at 55 ms.

It can be obtained that the rigid–flexible coupling collision between the finger rows of harvester and blueberry branch was transient, meaning that the harvesting collision phenomenon will occur regardless of the machine parameters of the harvester for blueberry harvesting operation.

Analysis of the branch vibration response surface shown in Figure 11 demonstrates that during the initial collision process from 0 s to 0.1 s, the branch vibration deflection deformation reached its maximum value of approximately 75 mm; with the end of the collision process, the branch vibration deflection deformation gradually decreased to 25 mm at 0.4 s; during the whole vibration process, the branch vibration deflection deformation gradually increased from the root to the end of the branch.

It also can be obtained that the blueberry branch near the root possessed a smaller vibration deformation, and the branch near the end possessed larger vibration deformation. Equation (3) reveals that the magnitude of fruit harvesting force was proportional to the degree of branch vibration deformation. In other words, the fruit harvesting force near the end of the branch was higher. Therefore, when pruning the blueberry plant, the branch from the root to the middle of the branch should be pruned off so that the fruit can grow as

fast as possible from the middle to the end of the branch, which can ensure the efficient harvesting of blueberry fruit and better realize the mutual matching between agronomy and agricultural machinery.

Analysis of the normal collision force curves and the tangential collision force curves of the branch shown in Figure 12a,b demonstrates that at 30 ms, the finger rows collided with the tree branch; as the collision process continued, the collision force fluctuated and increased, at 45 ms, the normal collision force reached its maximum value of about 90 N (at 45 ms, the tangential collision force reached its maximum value of about 55 N); subsequently, the collision force oscillated and decreased at 100 ms.

Then, the collision force started to gradually oscillate and decay under the interaction between the two, and the oscillation value tended to be 0 ms at 100 ms; comparing with the above conclusions, it can be analyzed that the normal collision force was larger than the tangential collision force.

As the normal collision force is the main factor of blueberry branch to produce fruit harvesting force, the design of the harvesting system should consider the spatial layout of the finger rows of harvester to ensure that the finger rows collide with the blueberry branch head-on as much as possible to reduce the tangential collision force and increase the normal collision force, so as to achieve an efficient collision between the finger rows and the plant.

Analysis of the harvesting force curves of the blueberry fruit shown in Figure 13 demonstrates that, at the moment of collision between the two interactions, the harvesting force of the blueberry fruit instantly reached a maximum value of about 4 N; with the end of the collision process, the harvesting force curve of the fruit continuously oscillated and decayed, the oscillation value of which decayed to 0 at 0.6 s.

### 3.1.2. Rigid–Flexible Coupling by ADAMS to Analyze Single-Point Collision

According to the setting parameters by ADAMS in Section 2.4.2, the fruit harvesting force curves, the normal collision force curves, and the tangential collision force curves of blueberry branch shown in Figures 12 and 13 were obtained by collision simulation analysis.

Figure 12a,b show that the normal collision force and the tangential collision force of MATLAB programming curves were greater than those of ADAMS simulation curves. This is because MATLAB programming only considers the branch deformation in the collision point area and the corresponding collision force. The blueberry branch established in ADAMS is a finite element model after meshing (one end of the branch is fixed and set as the root of the branch, while the other end is unconstrained and set as the tip of the branch), which is composed of multiple tiny cell bodies.

When the branch was subjected to collision force, the tiny cells in the collision point area absorbed collision energy and produced collision deformation and a collision force. Furthermore, the tiny cells in the area from the root of the blueberry branch to the collision points absorbed collision energy and produced corresponding collision deformation and a collision force. Since the tiny cells in the area from the root of the blueberry branch to the collision points absorbed collision energy, the collision energy absorbed by the tiny cells in the collision point area of ADAMS was smaller than that in the collision point area absorbed by the collision point of MATLAB. The corresponding MATLAB curves are shown in Figure 12a,b. The normal collision force and tangential collision force of MATLAB curves shown in Figure 12b were higher than the normal collision force and tangential collision force of the corresponding ADAMS curves.

Taking the data points of ADAMS curves and MATLAB curves shown in Figure 12 as the object of study, the relevance between the ADAMS curves and MATLAB curves was calculated using the F-test (joint hypotheses test), and the relevance was calculated as follows [18,19]:

$$F_{xy} = \frac{\sum_{i=1}^{m}(x_i - x)^2}{\sum_{j=1}^{n}(y_j - y)^2} \cdot \frac{y_{max}^2}{x_{max}^2} \tag{14}$$

where $x_i$ and $y_i$ are data points waiting to be tested; n is the number of data points; $\overline{x}$ and $\overline{y}$ are the mean values of data points waiting to be tested; $x_{max}$ and $y_{max}$ are the maximum values of data points waiting to be tested, $x_{max} = max\{|x_i|\}$, $y_{max} = max\{|y_i|\}$; $F_{xy}$ is the F-test value for data point $x_i$ and data point $y_i$.

The F-test values $F_{xy}$ of the ADAMS curves and MATLAB curves in Figure 12a,b, calculated using Equation (14), were 1.0168 and 1.0179, respectively, which were smaller than the corresponding standard values $F_0$: 3.042 and 3.052. Thus, the curves were not significantly different, indicating that the MATLAB programming curves and the ADAMS simulation curves by Equations (5) to (8) had a high degree of relevance and a good degree of fitting.

Then, we obtained that both the L-N nonlinear spring damping model and the improved Coulomb model introduced in Section 2.3 were suitable for blueberry harvesting collision studies and could analyze the blueberry collision harvesting process.

The fruit harvesting force curves in MATLAB and ADAMS in Figure 13 had the same variation tendency and similar values in the peak of force. Then, the curves were calculated by Equation (14): the F-test value $F_{xy}$ of the curves was 1.005, which was smaller than the corresponding standard value $F_0$: 3.089. Thus, the curves were not significantly different, indicating that the fruit harvesting force curves in MATLAB and ADAMS with the same parameter settings had a high degree of relevance and a good degree of fitting.

### 3.2. Analysis of Multi-Point Collision of Blueberry Plants and Harvesting Field Test

3.2.1. Multi-Point Collision Analysis with Rigid–Flexible Coupling under ADAMS

In ADAMS, the output rotational velocity of hydraulic motor of harvesting system was set to 200 r/min, and the machine velocity of harvester was set to 20 m/min. Then, the harvesting collision force and blueberry harvesting force were obtained, which are shown in Figure 14. The harvesting collision force measurement curves shown in Figure 14a,b were obtained from the collision measurement points between the harvester and the blueberry plant, while the fruit harvesting force measurement curves shown in Figure 14c,d were obtained from the fruit harvesting force measurement points between the harvester and the blueberry plant.

The only difference between Figure 14b,d and Figure 14a,c is that Figure 14a,c show the collision force and blueberry fruit harvesting force obtained when the blueberry plant was set as a flexible body under the consideration of the rigid–flexible coupling effect of blueberry harvesting and Figure 14b,d show the collision force and blueberry fruit harvesting force obtained when the blueberry plant was not transformed into a finite element model after meshing in ANSYS while directly imported into ADAMS through Mech/Pro to form a rigid body of blueberry plant without considering the rigid–flexible coupling effect of blueberry harvesting.

The mechanical parameters of the rigid body of the blueberry plant in Figure 14b,d and the collision connection between plant and the rigid body of the harvester were set in the same way as Figure 14a,c.

In these figures, the horizontal coordinate is the simulation time of the interaction between the harvester and the blueberry plant and the vertical coordinate was the harvesting collision force and the fruit harvesting force. As the relative position between the harvester and the blueberry plant was kept at a distance at the initial moment, the collision force and fruit harvesting force at the initial moment of the simulation was zero; as the simulation process proceeded, the harvester moved and came into contact with the blueberry plant, and the collision force and fruit harvesting force at the measurement point continuously changed; when the harvester was separated from the blueberry plant, the collision process between them ended and the simulation measurement curve was zero at this point. As the collision point between the harvester and the blueberry plant was a random contact point, the resulting collision force and fruit harvesting force were transient excitation forces, while the plant collision measurement point and fruit harvesting force measurement point set by the software were fixed points; hence, the obtained collision force measurement curve and

fruit harvesting force measurement curve were random curves with transient changes in the peak of the curves.

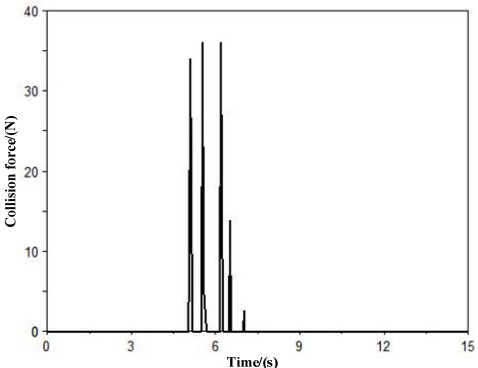

(**a**) Harvesting collision force under consideration of rigid–flexible coupling

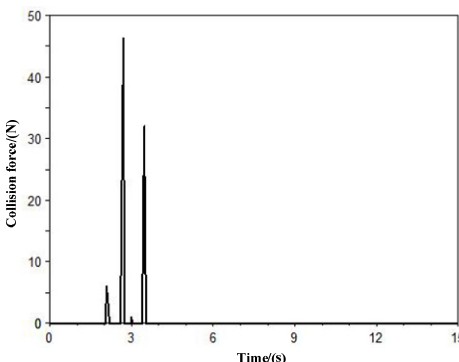

(**b**) Harvesting collision force without consideration of rigid–flexible coupling

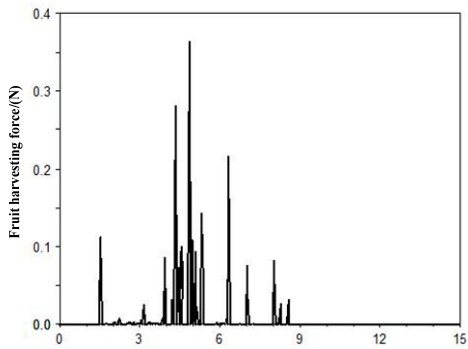

(**c**) Blueberry fruit harvesting force under consideration of rigid–flexible coupling

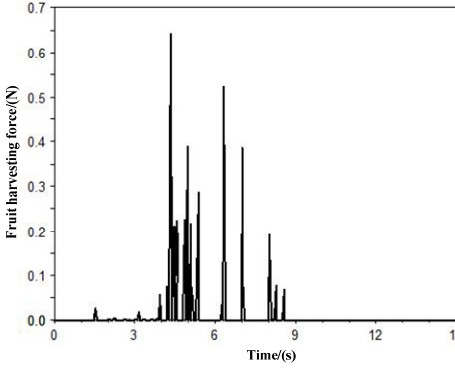

(**d**) Blueberry fruit harvesting force without consideration of rigid–flexible coupling

**Figure 14.** Comparative analysis of harvesting collision force between the harvester and the blueberry plants.

The force value of the peak on the curves was analyzed to obtain the number of mutual collisions. The maximum collision force and blueberry fruit harvesting force were obtained during the collision based on the specific value of the maximum peak of the curves, which were used as the collision force and blueberry fruit harvesting force during the interaction between the harvester and the blueberry plant, and the number of collisions, collision force, and blueberry harvesting force during the interaction between the two were analyzed to study the collision mechanism of blueberry harvesting.

By comparing Figure 13a–d, a certain deviation in the simulation results could be obtained when the blueberry plants were set as flexible and rigid bodies, respectively. A comparison of the horizontal coordinates of the curves shows that the time of peak on the curves changed, caused by the change in the location of the collision point due to the different settings of the plant. A comparison of the peak force values on the curves shows that the number of collisions was significantly higher when the blueberry plant was set as a flexible body than when the blueberry plant was set as a rigid body. A comparison of the vertical coordinates of the curves shows that the collision force and the fruit harvesting force were smaller when the blueberry plant was set as a flexible body than those when the plant was set as a rigid body. In other words, the values of collision force and the blueberry fruit harvesting force obtained, without taking the rigid–flexible coupling of the blueberry harvesting vibration into account, were greater. A comparison of the simulation results

shows that the rigid–flexible coupling collision between the harvester and the blueberry plant had an effect on both the simulation process and the simulation results.

　　In the process of rigid–flexible coupling with blueberry plants, since the finger rows were multi-excitation units (in the finger rows on both sides, the number of finger rows on each side was 15, the length of a single harvesting finger was 600 mm, the radius was 15 mm, and the distance between finger rows was 200 mm), which interacted with different positions of blueberry plants, different positions of the blueberry branch, as well as the contact points of the branch of different diameters to produce transient collision forces, the blueberry branch vibrated, thereby generating the corresponding fruit harvesting force. The following figure shows the influencing factors of collision force and fruit harvesting force during the harvesting operation.

　　When the machine velocity of the harvester was set to 40 m/min and the rotational velocity of the hydraulic motor of the harvesting system was set to different values in ADAMS, the collision harvesting force curves shown in Figure 15a–f were obtained according to the collision measurement points between the harvester and the blueberry plant.

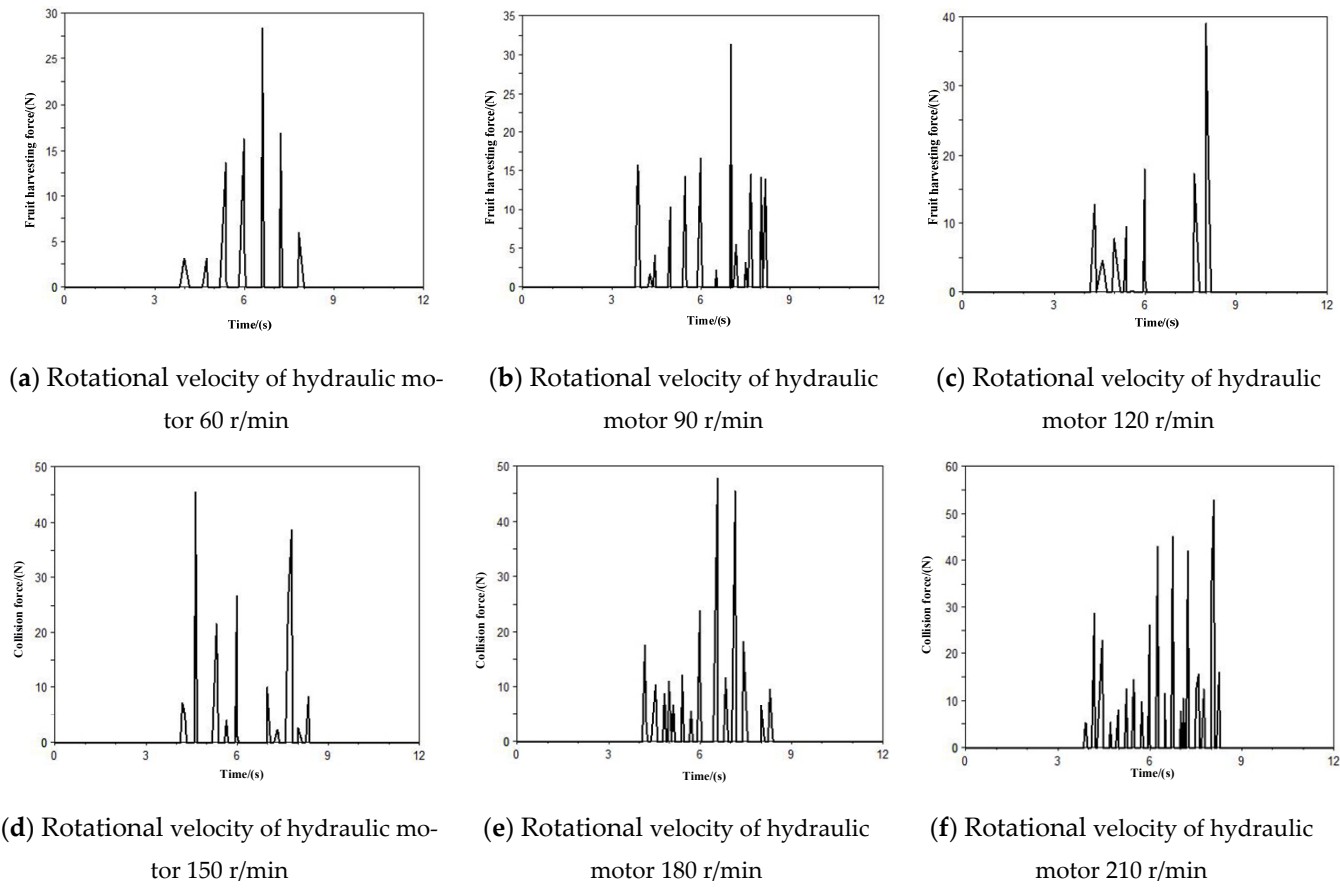

(**a**) Rotational velocity of hydraulic motor 60 r/min

(**b**) Rotational velocity of hydraulic motor 90 r/min

(**c**) Rotational velocity of hydraulic motor 120 r/min

(**d**) Rotational velocity of hydraulic motor 150 r/min

(**e**) Rotational velocity of hydraulic motor 180 r/min

(**f**) Rotational velocity of hydraulic motor 210 r/min

**Figure 15.** Analysis of the influence of rotational velocity on the harvesting collision force.

　　It can be seen from Figure 15 that with a continuous increase in the output rotational velocity of the hydraulic motor of the harvesting system, the transient collision force $\vec{F}_i(t)$ at the measurement point also increased, due to the fact that the increase in rotational velocity increased both the angular velocity $\omega$ and the angular acceleration of the finger rows when performing a reciprocating swing. The normal collision force $F_{ni}$ and the tangential collision force $F_{\tau i}$ of the blueberry branch at the collision point $A$ increased, according to Figure 4, and the collision force $\vec{F}_i(t)$ at the measurement point also increased, according to Equation (1). In other words, the collision force $\vec{F}_i(t)$ increased with an increase in the output rotational velocity of the hydraulic motor of the harvesting system.

When the machine velocity of the harvester was set to 40 m/min and the rotational velocity of the hydraulic motor of the harvesting system was set to different values in ADAMS, the fruit harvesting force curves were obtained according to the harvester and plant harvesting force measurement points, which are shown in Figure 16a–f.

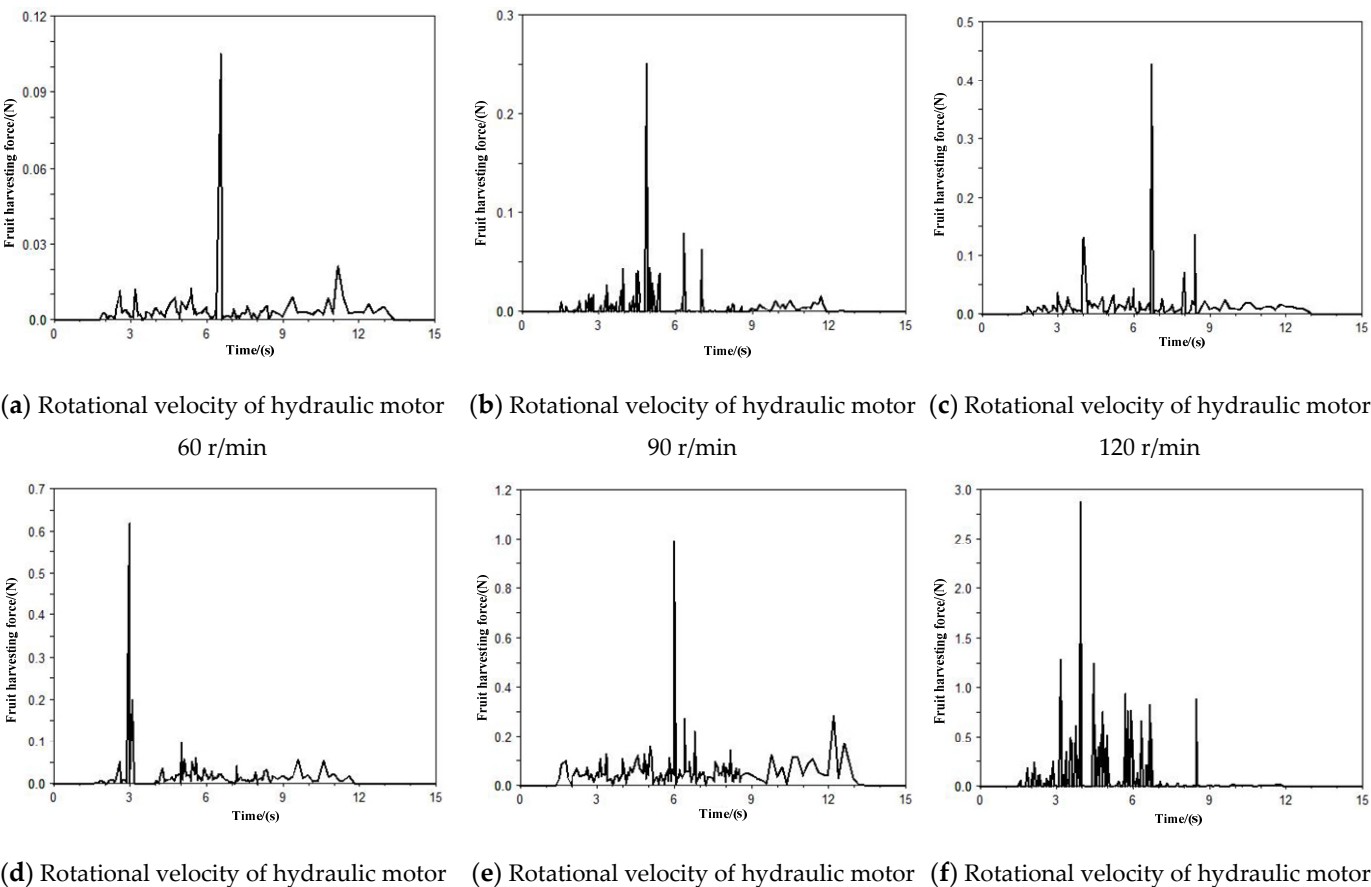

(**a**) Rotational velocity of hydraulic motor 60 r/min

(**b**) Rotational velocity of hydraulic motor 90 r/min

(**c**) Rotational velocity of hydraulic motor 120 r/min

(**d**) Rotational velocity of hydraulic motor 150 r/min

(**e**) Rotational velocity of hydraulic motor 180 r/min

(**f**) Rotational velocity of hydraulic motor 210 r/min

**Figure 16.** Analysis of the influence of rotational velocity on the harvesting force.

Figure 16 shows that with an increase in the output rotational velocity, the angular velocity $\omega$ of the finger rows on both sides increased when performing a reciprocating swing, the angular frequency $\omega_i$ of the vibration of the blueberry branch increased, and the fruit harvesting force $F_{maxi}^{f}(t)$ at the measurement point also increased, according to Equation (3).

When the output velocities of the hydraulic motor were 60 r/min and 90 r/min, respectively, the fruit harvesting forces were about 0.11 N and 0.25 N, i.e., both smaller than the bonding force between the ripe fruit and the branch $F_2 = 0.26 - 0.3\ N$ (based on the parameters in Equation (4)), and the value of which was small. The specific values corresponding to the maximum peak in the fruit harvesting force curves are shown in Figure 16a,b.

When the output velocities of the hydraulic motor were 120 r/min and 150 r/min, respectively, the fruit harvesting forces were about 0.4 N and 0.6 N, i.e., greater than the bonding force between the ripe fruit and the branch, but smaller than the bonding force between the raw fruit and the branch, meeting the harvesting conditions.

When the output velocities of the hydraulic motor were 180 r/min and 210 r/min, respectively, the fruit harvesting forces were about 1.1 N and 2.9 N, i.e., between the bonding force between the raw fruit and the branch $F_1 = 1.0 - 3.6\ N$ (based on notes on the parameters in Equation (4)), and the value of which was larger. The specific values

corresponding to the maximum peak in the fruit harvesting force curves are shown in Figure 16e,f.

In summary, the output rotational velocity range of the hydraulic motor of the harvesting system that meets the harvesting conditions ranged from 120 r/min to 150 r/min.

The maximum collision force shown in Figure 15a–f and the maximum fruit harvesting force shown in Figure 16a–f were used as data points for programming, and the velocity of the harvesting drive was set as the horizontal coordinate to obtain the corresponding collision force curves and fruit harvesting force curves, as shown in Figure 17.

The variation tendency of the curves in Figure 17 was observed and calculated by Equation (14). The F-test value $F_{xy}$ of the curves of collision force and the curves of fruit harvesting force was 3.67, which was smaller than the corresponding standard value $F_0$: 9.552. Thus, the curves were not significantly different, which indicates that the hydraulic motor's collision curves of force-rotational velocity and the hydraulic motor's curves of fruit harvesting force-rotational velocity had a high degree of relevance. In other words, both the collision force and the fruit harvesting force resulting from the interaction between the finger rows of the harvester and the blueberry branch were proportional to the output rotational velocity of the harvesting drive device.

When the rotational velocity of the hydraulic motor of the harvesting system was set to 120 r/min and the machine velocity of harvester was set to different values, the collision force curves were obtained according to the harvester and plant harvesting force measurement points, which are shown in Figure 18a–f. From these figures, it can be seen that with an increase in the machine velocity of the harvester, the interaction time between the finger rows and the blueberry branch decreased, the number of collisions decreased, the duration of continuous vibration of the blueberry branch decreased, and the collision force formed decreased.

The interaction collision process between the finger rows of the harvester and the blueberry branch shows that when the first collision occurred, the blueberry branch changed from stationary to a moving state under the action of the collision force. At the initial moment after the collision, the branch acquired the initial collision velocity, and the velocity direction was the action direction of the collision force; subsequently, under the action of the structural damping ξ of the branch, the collision velocity decayed rapidly to 0, and at this moment the branch produced deflection deformation. The branch began to move reversely under the action of the branch deformation force, and the velocity increased from 0, with a certain relative motion velocity to contact the finger rows, which went the opposite direction of motion. Then, the second collision occurred, which meant that the branch possessed a certain relative motion velocity, and the collision force generated by the second collision was greater than the first collision force. With the continuation of interaction times between the figure rows and the branch, the number of collisions gradually increased, the relative motion velocity of the branch gradually increased, and the collision force gradually increased. Conversely, with the increase in the machine velocity of the harvester, the interaction time between the two decreased, the number of collisions also gradually decreased, the relative movement velocity of the branch gradually decreased, and the collision force gradually decreased, forming the collision force variation curves in Figure 18a–f.

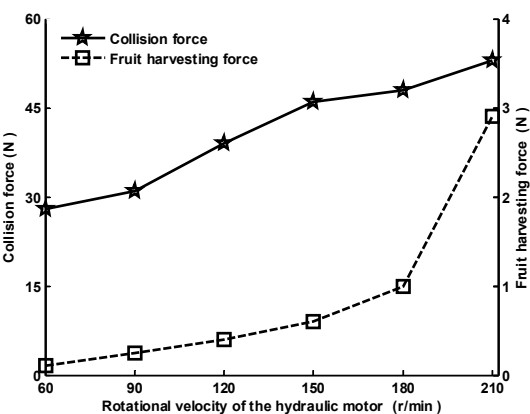

**Figure 17.** The variation curves of the influence in the rotational velocity of the hydraulic motor.

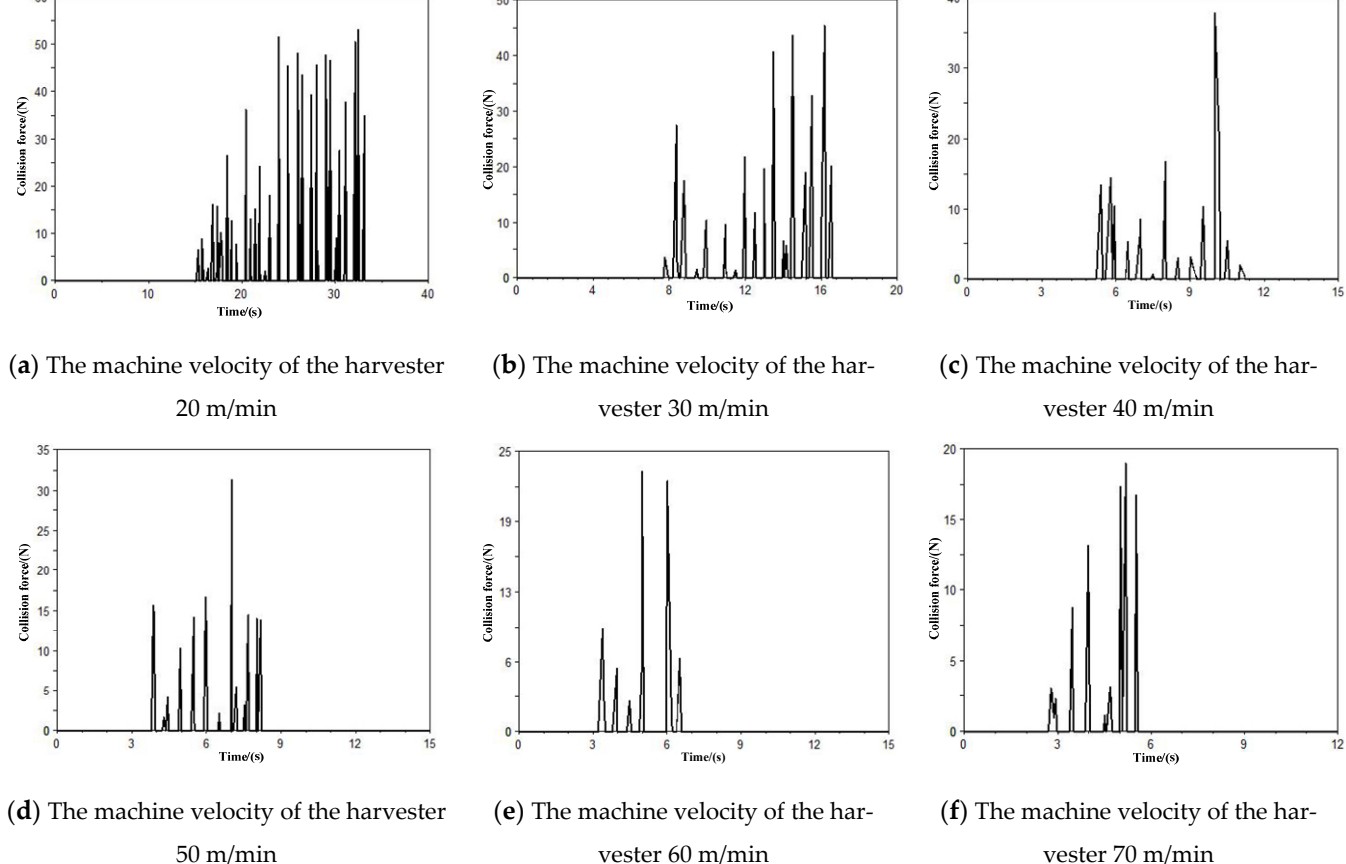

(**a**) The machine velocity of the harvester 20 m/min

(**b**) The machine velocity of the harvester 30 m/min

(**c**) The machine velocity of the harvester 40 m/min

(**d**) The machine velocity of the harvester 50 m/min

(**e**) The machine velocity of the harvester 60 m/min

(**f**) The machine velocity of the harvester 70 m/min

**Figure 18.** Analysis of the influence of the machine velocity on the harvesting collision force.

When the rotational velocity of the hydraulic motor of the harvesting system was set to 120 r/min and the machine velocity of harvester was set to different values, the fruit harvesting force curves were obtained according to the harvester and plant harvesting force measurement points, which are shown in Figure 19a–f.

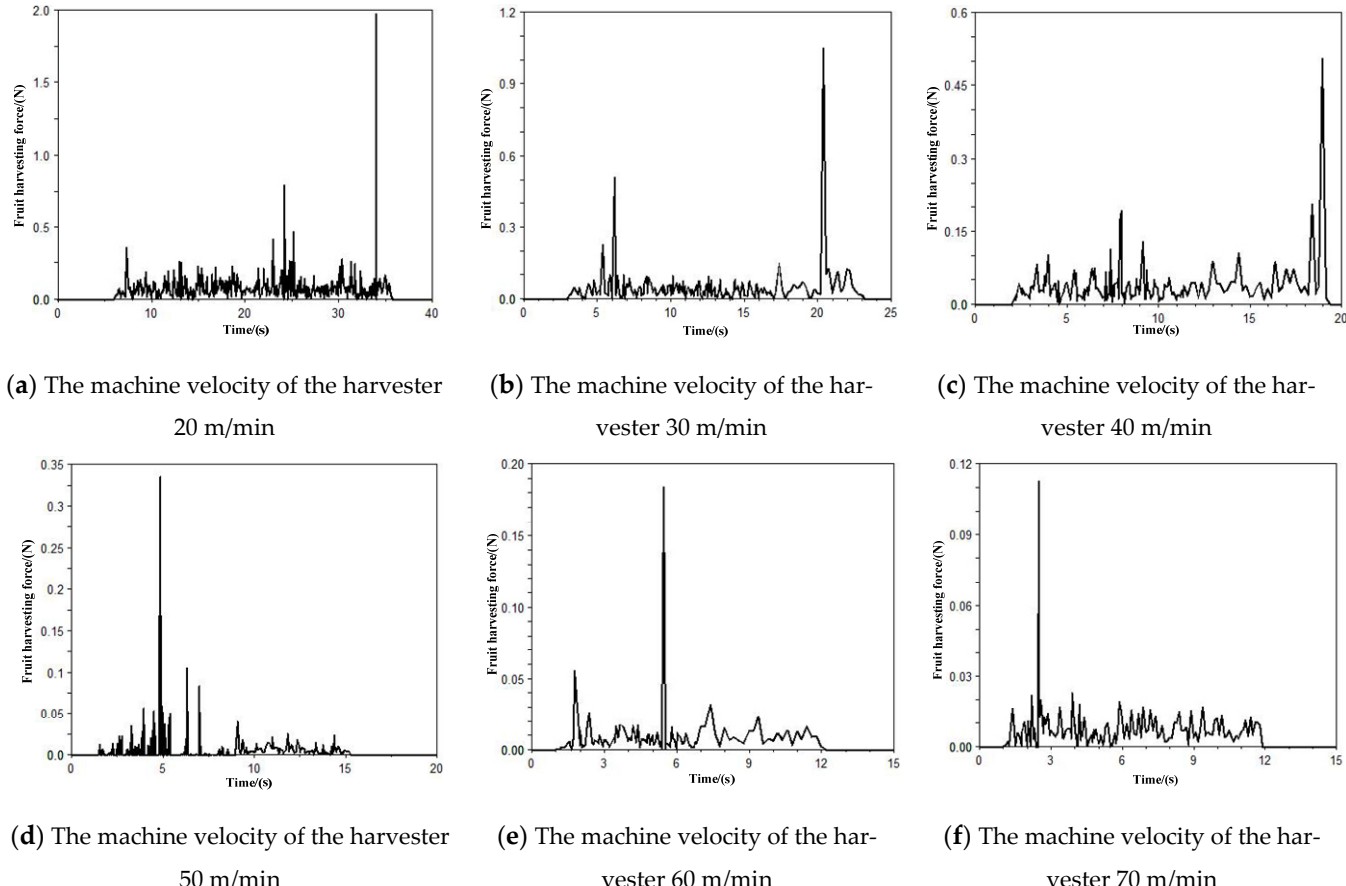

(**a**) The machine velocity of the harvester 20 m/min

(**b**) The machine velocity of the harvester 30 m/min

(**c**) The machine velocity of the harvester 40 m/min

(**d**) The machine velocity of the harvester 50 m/min

(**e**) The machine velocity of the harvester 60 m/min

(**f**) The machine velocity of the harvester 70 m/min

**Figure 19.** Analysis of the influence of the machine velocity on the blueberry fruit harvesting force.

The fruit harvesting force curves under different machine velocity conditions, when the output rotational velocity of the hydraulic motor of the harvesting system was 120 r/min, are shown in Figure 19a–f. It can be seen from the figure that as the machine velocity of the harvester increased, the fruit harvesting force gradually decreased. This is due to the fact that as the machine velocity of the harvester increased, the interaction time between the finger rows of the harvester and the blueberry plant gradually decreased, the number of collisions between the two interactions also gradually decreased, the relative kinematic velocity of the tree branch gradually reduced, the size of the collision force gradually decreased, and the resulting fruit harvesting force gradually decreased.

When the machine velocities of the harvester were 20 m/min and 30 m/min, the fruit harvesting forces were about 2.0 N and 1.1 N (the specific values corresponding to the maximum peak in the fruit harvesting force curves are shown in Figure 19a,b), i.e., between the bonding force between the raw fruit and the branch $F_1 = 1.0 - 3.6\ N$, and the value of which was larger. When the machine velocities of the harvester were 40 r/min and 50 r/min, respectively, the fruit harvesting forces were about 0.51 N and 0.34 N (the specific values corresponding to the maximum peak in the fruit harvesting force curves are shown in Figure 19c,d), i.e., greater than the bonding force between the ripe fruit and the branch but smaller than the bonding force between the raw fruit and the branch, meeting the harvesting conditions. When the machine velocities of the harvester were 60 r/min and 90 r/min, respectively, the fruit harvesting forces were about 0.18 N and 0.12 N (the specific values corresponding to the maximum peak in the fruit harvesting force curves are shown in Figure 19e,f), i.e., both smaller than the bonding force between the ripe fruit and the branch $F_2 = 0.26 - 0.3\ N$, and the value of which was small.

In other words, when the machine velocity of the harvester ranged from 40 m/min to 50 m/min, the fruit harvesting force was greater than the bonding force between the

ripe fruit and the branch but smaller than the bonding force between the raw fruit and the branch, meeting the harvesting conditions.

The maximum collision force shown in Figure 18a–f and the maximum fruit harvesting force shown in Figure 19a–f were taken as the programming data, and the machine velocity of the harvester was set as the horizontal coordinate to obtain the corresponding collision force curves and fruit harvesting force curves, as shown in Figure 20.

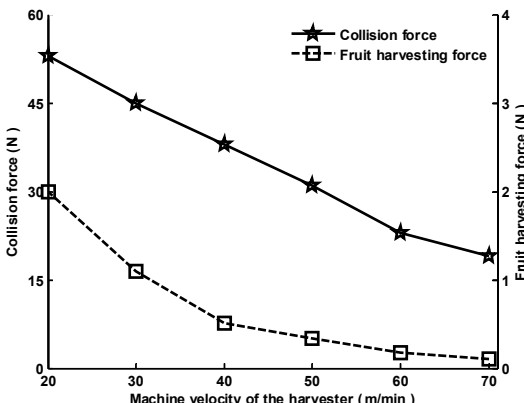

**Figure 20.** The variation curves of the influence of the machine velocity of the harvester.

The variation tendency of the curves in Figure 19 was observed and calculated by Equation (14). The F-test value $F_{xy}$ of the curves of collision force and the curves of fruit harvesting force in Figure 20 was 2.17, which was smaller than the corresponding standard value $F_0$: 9.552. Thus, the curves were not significantly different, which indicated that the curves of collision force machine velocity of the harvester and the curves of fruit harvesting force-machine velocity of harvester had a high degree of relevance. In other words, both the collision force and the fruit harvesting force resulting from the interaction between the finger rows of the harvester and the blueberry branch were inversely proportional to the machine velocity of the harvester.

In the process of blueberry harvesting, the fruit harvesting force of blueberries is produced by the collision force of the branch. The fruit harvesting force is the final deciding factor of whether the harvesting operation can realize "harvesting ripe fruit and leaving raw one".

The following results of the harvested fruit quality, combining Equation (4) and Figure 20, can be obtained. When the machine velocity of the harvesting machine was smaller than 40 m/min, the fruit harvesting force was small, which meant that some ripe fruit remained on the branch in the harvesting process and was not harvested. When the machine velocity was greater than 50 m/min, the fruit harvesting force was too large, which meant that some raw fruit was vibrated off. When the machine velocity was 40–50 m/min, the machine ensured better quality of harvested fruit, meeting the harvesting conditions. Similarly, when the rotational velocity of hydraulic motor was 120–150 r/min, the machine ensured better quality of harvested fruit, meeting the harvesting conditions.

It can be concluded that in order to achieve the harvesting purpose of "harvesting ripe fruit and leaving raw one", in the actual blueberry harvesting operation, it is appropriate to keep the machine velocity of the harvester and the rotational velocity of the hydraulic motor between 40 and 50 m/min and 120 and 150 r/min, respectively.

### 3.2.2. Analysis of Blueberry Harvesting Field Test

Table 4 shows that with the improved output rotational velocity of the hydraulic motor, the vibration angle frequency of the blueberry branch $\omega_i$ increased and the reciprocating swing angular velocity of the finger row $\omega$ increased. The law of centrifugal force states that when the normal collision harvesting force $F_{ni}$ increases, the vibrational deformation of the blueberry plant $A_i$ increases. Equation (3) shows that the fruit harvesting force

$F^f_{maxi}(t)$ was proportional to the vibrational deformation of the plant $A_i$ and the vibrational angular frequency of the branch $\omega_i$, so the fruit harvesting force $F^f_{maxi}(t)$ also increased and more blueberries were dropped by the finger rows. The harvesting efficiency, harvested raw fruit rate, and harvested damaged fruit rate also increased. In other words, as the output rotational velocity of the hydraulic motor in the harvesting system $\omega$ increased, the collision force of the blueberry branch $\overrightarrow{F}_i(t)$ and the fruit harvesting force $F^f_{maxi}(t)$ increased, similar to the harvesting efficiency of the machine, the harvesting rate of raw fruit, and the harvesting rate of damaged fruit.

**Table 4.** Results of the blueberry harvesting field test.

| Sequence Number | Output Rotational Velocity (r/min) | Machine Velocity (m/min) | Harvesting Efficiency (kg/min) | Harvesting Rate of Raw Fruit (%) | Harvesting Rate of Damaged Fruit (%) |
|---|---|---|---|---|---|
| 1 | 70 | 28 | 3.5 | 3.3 | 3.7 |
| 2 | 130 | 28 | 4.2 | 4.8 | 4.2 |
| 3 | 190 | 28 | 4.9 | 5.4 | 4.7 |
| 4 | 120 | 28 | 3.9 | 4.6 | 4.1 |
| 5 | 120 | 45 | 5.1 | 2.9 | 3.6 |
| 6 | 120 | 68 | 4.8 | 2.6 | 3.4 |

Table 4 shows that as the machine velocity of the harvester increased, the interaction time between the finger rows of the harvester and the blueberry branch was shortened, the number of mutual collisions between the two decreased, the kinematic velocity of the branch also gradually decreased, and the collision force was gradually reduced, resulting in lower blueberry fruit harvesting force and a reduced amount of fruit being vibrated off.

In addition, because the blueberry branch is flexible body, a single collision can only vibrate off the blueberry fruit growing on the end of the colliding branch and the fruit around the collision points. Thus, as the number of collisions and the total number of collision points all decreased, the amount of blueberry fruit vibrated by the harvester finger rows decreased.

Equation (3) reveals that the angular frequency of vibration of the blueberry branch $\omega_i$ decreased and the harvesting force of the blueberry fruit $F^f_{maxi}(t)$ decreased, resulting in a reduction in the harvesting efficiency of the machine and a reduction in the harvesting rate of raw fruit and the rate of harvested damaged fruit. This means that as the machine velocity of the harvester increased, the fruit harvesting force $F^f_{maxi}(t)$, the harvesting efficiency of the machine, the harvesting rate of raw fruit, and the harvesting rate of damaged fruit all decreased.

By combining the testing data of the above six groups, it became clear that when the machine velocity of the harvester was set to 45 m/min and the output rotational velocity of the hydraulic motor in the harvesting system was set to 130 r/min, the harvesting efficiency and the quality of the harvested fruit of the machine reached the best at this time, as the harvesting efficiency was 5.1 kg/min, the harvesting rate of raw fruit was 2.9%, and the rate of harvested damaged fruit was 3.6%.

## 4. Conclusions

Firstly, the collision harvesting mechanism of blueberry fruit was investigated, the effect of the variation of fruit harvesting force on the type of fruit harvested was analyzed, and the vibration harvesting conditions of ripe blueberry fruit were determined. The L-N non-linear spring damping model and the modified Coulomb friction model were used to construct the collision harvesting model with rigid–flexible coupling between the harvester and the blueberry plant, the ADAMS simulation results were compared, and the MATALAB software was programmed to verify the established blueberry collision harvesting model.

Secondly, ANSYS software and ADAMS software were integrated to simulate the collision harvesting process of blueberry fruit. The simulation results were compared and found that in the harvesting process, the rigid–flexible coupling collision factor had an influence on the collision force and fruit harvesting force. The influence of the working parameters of the machine on the harvesting process was analyzed, revealing that the collision force and fruit harvesting force were inversely proportional to the machine velocity of the harvester, but positively proportional to the output rotational velocity of the hydraulic motor of the harvesting system. Therefore, the better working parameters of the machine to meet the harvesting conditions were an output rotational velocity of the harvesting drive ranging from 120 r/min to 150 r/min and a machine velocity of the harvester ranging from 40 m/min to 50 m/min.

Thirdly, a self-propelled blueberry harvester was used and combined with the single-factor method to carry out experimental research on blueberry fruit harvesting, and the following conclusions were obtained: as the output rotational velocity of the hydraulic motor of the harvesting system increased, the collision force of the blueberry branch and the fruit harvesting force all increased, and the harvesting efficiency, the harvesting rate of raw fruit, and the harvesting rate of damaged fruit also increased; however, as the machine velocity of the harvester increased, the collision force of the blueberry branch and the fruit harvesting force all decreased, and the harvesting efficiency, the harvesting rate of raw fruit, and the harvesting rate of damaged fruit also decreased. When the machine velocity of the harvester was 45 m/min and the output rotational velocity of the hydraulic motor of the harvesting system was 130 r/min, the harvesting efficiency and the quality of fruit harvested of the machine reached the optimal level, as the harvesting efficiency was 5.1 kg/min, the harvesting rate of raw fruit was 2.9%, and the harvesting rate of damaged fruit was 3.6%.

Finally, in light of the above analysis, this paper focused on the collision process between the harvesting device of the harvester and the blueberry plant after manual pruning under the interaction of rigid–flexible coupling. In the harvesting process, "collision" is the core problem. Most of the literature has focused on rigid body–rigid body collision, though few authors have studied the rigid–flexible coupling collision mechanism in the field of berry harvesting.

This research can lay the preliminary theoretical foundation for the analysis of the blueberry harvesting mechanism and accelerate the process of mechanical blueberry harvesting in China. The most important point is that the research results can provide a theoretical reference for the harvesting of other similar berry shrubs.

**Author Contributions:** Conceptualization, H.W. and X.L.; methodology, H.W.; software, H.W.; validation, F.X. and L.S.; formal analysis, X.L.; investigation, F.X.; data curation, L.S.; writing—original draft preparation, X.L.; writing—review and editing, X.L.; visualization, L.S.; supervision, H.W.; project administration, H.W.; funding acquisition, H.W. All authors have read and agreed to the published version of the manuscript.

**Funding:** This research was funded by the Science and Technology Department of Heilongjiang Province of China (grant number: LH2020C047), the China Postdoctoral Science Foundation (grant number: 2019T120248) and the Northeast Forestry University (grant number: 2572022DP01).

**Institutional Review Board Statement:** Not applicable.

**Informed Consent Statement:** Not applicable.

**Data Availability Statement:** The data presented in this study are available on request from the corresponding author.

**Acknowledgments:** We would like to thank the State Key Laboratory of Tree Genetics and Breeding of Northeast Forestry University for their robust support in the field test of this research. We would also like to thank Li Zhipeng and Guo Yanling for their guidance and advice on the project.

**Conflicts of Interest:** The authors declare no conflict of interest.

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
