# Peer review of "Analysis and Testing of Rigid–Flexible Coupling Collision Harvesting Processes in Blueberry Plants"

_agriculture, doi:10.3390/agriculture12111900_

Round 1

Reviewer 1 Report

1.     Please include the full definition of L-N in the abstract.

2.     The introduction of the L-N nonlinear spring damping model and the improved Coulomb model used in the previous studies are not included in the literature review. Please include a paragraph of these models that were used in the previous works on agricultural research.

3.     Please capitalize the ‘figure’ in the text (page #9, line#317,318).

4.     The discrepancy analysis is not included in the results, as presented in Figure 9, and discusses why Matlab overpredicts the ADAMS.

5.     The capital letter error of unit Mpa (Page 12, line#389) should be MPa.

6.     The dimension of the blueberry plant model is not mentioned in the manuscript.

7.     Please include the coefficient of determination in Figures 15 and 18 to show the correlation between the dependent and independent parameters.

Author Response

Dear Reviewer:

We are delighted to receive your detailed feedback, which has been of great benefit to us. In response to your comments, we have responded and improved each of them.

In order to give a more accurate and detailed answer, I have organized the answer in a pdf and sent it to you as an attachment, please check it!

We deeply appreciate your comments of our manuscript, which helps to improve the quality of our paper. If you have any queries, please don’t hesitate to contact me at the address below. Thank you again for reviewing our manuscript!

Best regards,

Sincerely Yours,

Dr. Haibin Wang School of Engineering and Technology, Northeast Forestry University, China

whb_nefu@nefu.edu.cn

Reviewer 2 Report

The manuscript has some flaws that are given in the attached file

Author Response

(The authors gave the same response as above.)

Reviewer 3 Report

Comments to the authors:

Dear author(s):

First of all, thank you for your submission. I appreciate the effort you put into your study and acknowledge that it presents great potential. However, the text is poorly written, the study’s novelty and contribution to science were never clearly explained in the text, and the methodology contains flaws that should be fixed before publication. Please find below detailed comments:

What is the justification for this study? As far as I understood, the literature is broad in terms of models for blueberry harvesting. You mentioned that studies for China specifically are limited. However, I wonder if any particularities about blueberries cultivated in China or harvesting operations in China warrant another research study. Furthermore, the novelty of your study is not clear to me. What have you done in this study that differs from what’s available in the literature?

Abstract: The abstract lacks a proper introduction and a justification for the study.

Methodology: It was not clear the rationale behind validating your model. As far as I understood, you used ADAMS (a simulation software) to validate the results of MATLAB (another simulation software). What’s the purpose of this? The validation of the results should be carried out in the field with actual machines. Furthermore, this section lacks enough detail to allow your study to be replicated. For example, which tractor model did you use in your simulation? What was the moisture content of the fruits when they were harvested? Which parameters were fixed, and which parameters were variable in your simulation?

In addition, I recommend adding tables to summarize all parameters that were considered (for example, equations 1-16), their units, and the sources from which you retrieved values.

Results: I was shocked to see that the first four pages of your results are material pertinent to the Material and Methods section. Furthermore, it is not clear how some of the results were obtained. For example, in lines 521-522, you state, “When the walking speed of the harvester is 20m/min and 30m/min, the fruit harvesting force is about 1.2N and 2.0N.” Where did those values come from? It is not clear to me when looking at Figures 17a-17f.

Blueberry harvesting testing: This experiment was never mentioned in the Material and Methods section. It was unclear what machine parameters were tested and how the experimental design was selected.

Miscellaneous comments:

- Please review the text both in the Scientific and English languages. Some of the sentences are confusing and difficult to understand. For example, when talking about the “output rotational speed of the harvesting machine drive device,” you mention the values 120-150 r/min. What is r? If r stands for “radians,” it should be written as “rad/min.” In another example, you mention the “walking speed” of the harvester. Although it makes sense, that terminology is unusual for scientific writing. Instead, you could use the term “machine speed.”

- On lines 34 and 37, you need a reference for your sentences.

- What’s “mature picking” (line 42)

- Line 60: “… He Peizhuang, etc.” – why are you using “etc.?”

- Line 67: “harvesting power of blueberry fruits” – This is an example of corrections needed in the English language. I know it was not your intention, but the way it is written, it sounds like blueberry fruits are the ones performing the harvesting operation.

- Line 95: You state, “Not much research has been done in China,” and right after, you add four references (24-28). This is at least controversial.

- Line 309: “the speed of the branch collision was set to 0 m/s.” If the speed was 0, then how did the collision happen?

- Line 375: “model possessed a good fit with the simulation curve of ADAMS.” Based on what metrics did you make this conclusion? How was the goodness of fitness assessed?

- Line 382: “The growth pattern of blueberry plant was analyzed….” I am not sure if you actually analyzed the growth pattern of blueberry plants or how this was carried out.

- Line 553: “self-developed” – I believe you meant “self-propelled.”

- Line 569: I spent 20 minutes searching on Google and Google Scholar and couldn’t find any material related to the blueberry species “Blue Fung.” Could you please provide more information?

- Line 575 (Table 1): What are the numerical values presented in this table? This isn’t very clear. Furthermore, the units of the machine parameters are presented differently from the rest of the text (e.g., r.min-1 vs. r/min) – please standardize it.

These are just some of my observations. I recommend that the author(s) perform a full review of the text and work to improve the significant flaws of their work before re-submitting it again for publication.

Author Response

(The authors gave the same response as above.)

Round 2

Reviewer 3 Report

Comments to the authors:

Dear author(s):

Thank you for your response letter and updated manuscript. I appreciate the effort you put into it. While I recognize that you worked on the text, I still think it is hard to follow and confusing. You should work to improve its readability before the manuscript can be accepted for publication. Please find below detailed comments:

In my previous comment I asked what the justification for your study was, and what’s its novelty. You provided a comprehensive explanation in your response letter. Nevertheless, I think you could better explain it in the Introduction of your manuscript. For instance:

- You mentioned differences in planting agronomy between China and other countries. Please further elaborate. What are these differences? Why can’t machines developed in other countries be used in China?

Introduction

- I really appreciated that you clearly stated the novelty of your study (rigid-body – flexible-body interaction). I think that this should be the main point of your justification and your introduction should be developed around this topic. With that said, I suggest you rewrite the Introduction to make rigid-body – flexible-body interaction the central point of it. In addition, I would remove several parts that are not relevant to the topic and only make your Introduction long and boring. For example:

“The USA was the first country to start researching blueberry harvesters and has developed manual harvesting technology. Researchers have done a lot of relevant study to improve mechanised harvesting techniques and have achieved a wealth of corresponding research results. The two types of self-propelled rotary blueberry harvesters and swing harvesters that have been developed and put into production in the USA belong to the category of contact harvesters [1]. The contact harvester wrought by using the finger rows at the end of the actuator to vibrate and tap the target blueberry plant at a certain frequency in order to harvest the fallen blueberries. In many countries, the technology for mechanized blueberry harvesting has matured to a certain extent and can be widely applied to mechanized harvesting of high and low bush blueberries.

In a preliminary study of mechanized blueberry harvesting, Luca et al. in Italy compared a prototype machine and a commercial harvester (Easy Harvester®) with manual harvesting and pointed out that the labor costs of mechanized harvesting was greatly reduced. It was also significant to take into account that in the gradual transition from manual harvesting to mechanized harvesting, it was necessary to transform the farming methods and the operation of the packaging workshop, such as new crop varieties, field configurations and cultivation techniques [6].”

- Furthermore, you should clearly state how the outcomes of your study can be used in real-life applications. For example, several your results/discussion analyze the force of collision to harvest the blueberries. How exactly this information can be used by a farmer, or even a machine manufacturer? On Figure 9, you have a graph that shows the relationship between “X-axis”, “Y-axis”, and “Collision time”. How can this information be used by stakeholders? The same comment is valid for all other results. Please reflect and explain how the results translate into practical applications.

Material and Methods

- I am having a hard time understanding the steps of your study. I ask you to please include a sub-section right in the beginning of the M&M to provide an overview of your study. In this sub-section, please write one paragraph explain how the study was carried out. Mention that the first step consisted of reviewing the literature to lay out the theoretical equations that describe interaction between rigid and flexible body collisions. The second step was to implement these equations on Matlab; the third step was to implement these equations on ADAMS (here you can mention why you used both software); and the last step was to conduct field tests. You can even include a picture of a diagram showing the various steps of your study.

- After reading your response letter, it sort of makes sense to me why you used two different simulation software (ADAMS and Matlab). However, I do not think that it is appropriate to conclude that your simulations are realistic just because both software provided similar simulation outcomes. The only way of reaching this conclusion is by testing it in the field. I suggest you clearly explain your rationale behind using two software and emphasize that it is difficult to measure tangential force data; for this reason, you opted for testing your simulations in two different and independent software. It is ok if your methodology is not “bullet proof” and your study presents some limitations, as long as you state it.

- Table 1: What’s serial number? Maybe “ID” would be a better term here.

- Table 2: Please add units to the variables.

Miscellaneous comments:

- Revolutions per minute is usually abbreviated as “RPM”

- Line 111: “etc” – When citing a study performed by several scholars you should write the name of the first author followed by “et al.”. For example, “Guo Yanling and Bao Yudong, etc.” should be written as “Yanling et al.”.

- Line 518: The correlation coefficient is one out of several metrics that can evaluate if two independent variables present a linear relationship with each other. There are better metrics to evaluate if two models are the same (i.e., F-test). Moreover, under no circumstances one should conclude that the model presents “good fit” solely based on the coefficient of correlation. I suggest you use an F-test to test if the two models are the same (i.e., if they are correlate) and simply remove the part related to the goodness of fit.

-Line 43: What do you classify as “simple harvesting tools”? I would definitely not consider a harvester as a simple tool.

- Line 89: You need a reference here. Based on which data did you conclude that manual harvesting is the main method of harvesting blueberries?

- Line 91: “a lot of” is not appropriate scientific language for a manuscript. Consider changing to "significant”.

- Line 93: there’s no previous mention of the abbreviation “USA”. Please write it fully for the time.

- Line 94: “manual harvesting technology”; are you talking about harvesters? Those are not considered manual harvesting technology.

Lines 110-218: This entire section needs to be revised. There’s absolutely no need to spend 118 lines describing the work of others. Briefly mention that other studies have evaluated/developed blueberry harvesting models, but none has evaluated rigid-body – flexible-body collisions. Your study is the first to propose this novel approach.

Line 674: “The Coulometric”. Did you mean the coulometric equation? It feels like there’s something missing here…

Line 754: This is still part of the methodology. It doesn’t make sense to include conclusions here. Change the wording.

Line 782: This is not the first time you write “r/min”. Please move this Note to the part where you first mention “r/min”.

Line 798: What is a harvesting operation of a single blueberry? I believe there’s a writing error here. Perhaps you could describe that you set up your simulations to last 30 s each.

Line 801: What is(are) the response variable(s) here? It is unclear what mean values you’re referring to.

Line 817: “single blueberry”

Line 829: Throughout the text, you added a blank space when referring to units. Here, there’s no space between “30” and “ms”. The same observation is valid for line 834. Please standardize it.

Line 874: “i0n”

Lines 876-905: This single sentence is 29 lines long. I cannot stay focused and follow the flow of information with such a long sentence. I honestly doubt other people can. Please rewrite this sentence and break it down to improve the flow of information. Also, there are several other sentences in the text that need to be rewritten. Please work with all authors and/or with an editor to catch those sentences and improve their readability.

Line 910: Per Figure 12, the forces differ minimally. Moreover, there’s no background information in the text that clarifies if the force values are high or low. Does it really matter if Matlab calculates a force of 4.1 N while ADAMS calculated a force of 3.8 N? What is the real impact of this 0.3 N-difference?

Line 922: Please read my previous comment about correlation coefficient.

Line 1017: “walks”. Please change this word.

Line 1022: “contact” instead of “con-tact”.

Line 1027: What is the peak number of the curve? I believe you meant “the peak force value” observed in the graph.

Lines 1232-1246. Another long sentence.

Line 1567: Why is there a break in the sentence?

Line 1896 – Table 3: “Data of the test results”. Please add a descriptive title to this table.

Author Response

Dear Reviewer:

We are pretty delighted to receive your feedback on our revised manuscript again, your comments are very important for our research. After the first round of revisions, the level of our manuscript has been substantially improved by the suggestions of the three reviewers, including you. According to the second round of revisions you have given us, we have carefully addressed your comments and revised the manuscript accordingly.

The revised parts of manuscript were marked and screenshotted in the response letter. In order to provide more accurate and detailed answers, we have compiled the answers into a separate PDF file and submitted it as an attachment. Please review them!

Best regards,

Sincerely Yours,

Dr. Haibin Wang, School of Engineering and Technology, Northeast Forestry University, China

whb_nefu@nefu.edu.cn
